# EFFICIENT QUANTIZATION-AWARE TRAINING WITH ADAPTIVE CORESET SELECTION

## ABSTRACT

The expanding model size and computation of deep neural networks (DNNs) have increased the demand for efficient model deployment methods. Quantization-aware training (QAT) is a representative model compression method to leverage redundancy in weights and activations. However, most existing QAT methods require end-to-end training on the entire dataset, which suffers from long training time and high energy costs. Coreset selection, aiming to improve data efficiency utilizing the redundancy of training data, has also been widely used for efficient full-precision training. In this work, we propose a new angle through the coreset selection to improve the training efficiency of QAT. We propose two metrics based on analysis of loss and gradient of quantized weights: error vector score and disagreement score, to quantify the importance of each sample during training. Guided by these two metrics of importance, we proposed a quantization-aware adaptive coreset selection (ACS) method to select the data for the current training epoch. We evaluate our method on various networks (ResNet-18, MobileNetV2), datasets(CIFAR-100, ImageNet-1K), and under different quantization settings. Compared with previous coreset selection methods, our method significantly improves QAT performance with different dataset fractions. Our method can achieve an accuracy of 68.39% of 4-bit quantized ResNet-18 on the ImageNet-1K dataset with only a 10% subset, which has an absolute gain of 4.24% compared to the random baseline.

## 1 INTRODUCTION

Deep learning models have achieved remarkable achievements across various applications, including computer vision (Krizhevsky et al., 2012; He et al., 2016; Tan & Le, 2019) and natural language processing (Kenton & Toutanova, 2019; Yang et al., 2019; Conneau & Lample, 2019). The outstanding performance of these models can be attributed to their large number of parameters and the availability of large-scale extensive training datasets. For example, Swin-L (Liu et al., 2021) of input size $224^2$ has a total number of parameters of 197M with FLOPs of 34.5G. The language model GPT-3 (Brown et al., 2020) boasts astonishing 175 billion parameters and the pre-training is performed on the dataset with more than 410 billion tokens. The high latency, large model size, and large training set scale have become the most significant challenges for the training and deployment of deep learning models, especially on edge devices with computation limitation and storage constraints.

To address these challenges and enable the effective deployment of deep learning models, many model compression methods have been proposed recently. These model compression techniques include quantization (Zhou et al., 2016; Choi et al., 2018; Esser et al., 2020; Bhalgat et al., 2020), pruning (Liu et al., 2017; 2018; Molchanov et al., 2019; Liu et al., 2019), knowledge distillation (Hinton et al., 2015; Park et al., 2019; Shen & Xing, 2021), and compact network design (Howard et al., 2017; Pham et al., 2018). Among the aforementioned methods, quantization methods have been the most widely adopted because they have the advantage of promising hardware affinity across different architectures (Judd et al., 2016; Jouppi et al., 2017; Sharma et al., 2018). To minimize the performance gap between the quantized and full-precision models, quantization-aware training (QAT) is often utilized. Although QAT has improved the inference efficiency of the target model, it is computation-intensive and requires more time than full-precision training. Thus, training efficiency is important when QAT is applied to different networks and datasets.

Coreset selection techniques aim to mitigate the high training cost and improve data efficiency. Specifically, coreset selection methods leverage the redundancy in training datasets and select the most informative data to build a coreset for training. Previous methods select data based on feature (Agarwal

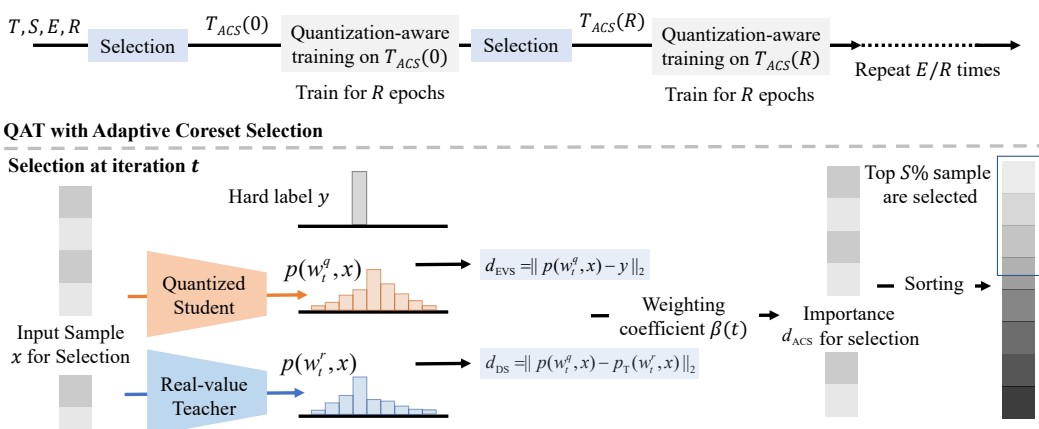

Figure 1: An overview of the adaptive Coreset Selection (ACS) for Quantization-aware training.

et al., 2020; Sener & Savarese, 2018), error (Paul et al., 2021), gradient (Killamsetty et al., 2021a), and decision boundary (Margatina et al., 2021) can achieve notable efficiency improvement for full-precision training, but its effectiveness on QAT has not been explored before. To utilize coreset selection methods to improve the efficiency of QAT, the characteristic of quantized weights must be considered in the design of methods. Existing full-precision coreset selection methods are not designed for quantization, and severe computation overhead during selection may hinder the application of QAT. A coreset selection tailored for QAT is needed for efficient training.

In this paper, we start by analyzing the impact of removing one specific sample from the training set during QAT and identify that the error vector score (EVS) is a good theoretical approximation for the importance of each sample. Based on the common practice of utilizing knowledge distillation during QAT, we also propose the disagreement score (DS) measuring the prediction gap between the quantized model and full-precision models. Based on these two metrics, we propose a fully quantization-aware adaptive coreset selection (ACS) method to select training samples that fit the current training objective considering current epochs, ACS, and DS. An overview of the proposed coreset selection method tailored for QAT is shown in Fig. 1.

We demonstrate the superiority of our ACS in terms of effectiveness and efficiency on different networks (ResNet-18, MobileNet-V2), datasets (CIFAR-100, ImageNet-1K), and quantization settings. For 2-bit weights-only quantization of MobileNetV2 on the CIFAR-100 dataset, QAT based on our ACS can achieve a mean accuracy of 67.19% with only 50% training data used for training per epoch. For 4-bit quantization of ResNet-18 on the ImageNet-1K dataset, our ACS can achieve top-1 accuracy of 68.39% compared to the 64.15% of random selection when only 10% training data is selected for the training of every epoch. Our proposed ACS can help accelerate the QAT without substantial accuracy degradation. In summary, our contribution can be summarized as follows:

- We are the first to investigate the training data redundancy of quantization-aware training from the perspective of coreset selection and find that importance varies among samples.
- We propose two metrics, error vector score (EVS) and disagreement score (DS), to quantify the importance of each sample to the QAT based on theoretical analysis of loss gradient.
- We propose a quantization-aware Adaptive Coreset Selection (ACS) method, which adaptively selects informative samples that fit current training stages based on EVS and DS.
- We verify our method ACS on different network architectures, datasets, and quantization settings. Compared with previous coreset selection methods, our ACS for QAT can significantly improve training efficiency with lower performance degradation.

## 2 RELATED WORK

**Quantization** Quantization methods are a powerful tool to improve the efficiency of model inference. The core insight is replacing full-precision weights and activations with lower-precision representation. Based on the characteristics of quantization intervals, these methods can be categorized into uniform and non-uniform quantization. While uniform quantization (Zhou et al.,

2016; Choi et al., 2018; Esser et al., 2020) with uniform interval are more hardware-friendly and efficient, Non-uniform quantization (Miyashita et al., 2016; Zhang et al., 2018; Li et al., 2019b), due to the flexibility of representation, it can minimize the quantization error and achieve better performance than uniform schemes. In addition, the quantization methods can also be classified as quantization-aware training (QAT) (Zhou et al., 2016; Esser et al., 2020; Bhalgat et al., 2020) and post-training quantization (PTQ) (Nagel et al., 2020; Fang et al., 2020; Wang et al., 2020) based on whether to retrain a model with quantized weights and activations or start with a pre-trained model and directly quantize it without extensive training. QAT can better retrain the performance and work with extremely low-precision quantization such as 2-bit, while PTQ can only achieve 8-bit and 6-bit quantization. However, QAT is more computation-intensive due to the extra training. In this work, we focus more on the efficiency of QAT methods.

**Coreset Selection**  Coreset selection targets improving the data efficiency by identifying the informative training samples. The essential part of coreset selection is quantifying each sample's importance. Previous works can be mainly classified into the following categories: **Geometry-based Methods:** These methods assume that the samples close to each other in the feature space similarly influence the training. These redundant data points should be removed to improve efficiency. Some representative works include Contextual diversity (CD) (Agarwal et al., 2020), k-Center-Greedy (Sener & Savarese, 2018), Sorscher et al. (2022), Moderate Coreset (Xia et al., 2023), and Herding (Welling, 2009). **Decision Boundary-based Methods:** These methods select samples distributed near the decision boundary. These samples are difficult for the model to separate. The works that fall in this method include Adversarial Deepfool (Ducoffe & Precioso, 2018), and Contrastive Active Learning (CAL) (Margatina et al., 2021). **Gradient/Error-based Methods:** These methods assume the training samples are more informative if they contribute more to the error or loss during the training. The contribution to the loss can be approximated with the gradient on the sample. Forgetting events (Toneva et al., 2019) are also an indicator of error during training. This category includes works such as EL2N (Paul et al., 2021), CRAIG (Mirzasoleiman et al., 2020), GradMatch (Killamsetty et al., 2021a), and AdaCore (Pooladzandi et al., 2022). **Optimization-based Methods:** These methods formulate the coreset selection as a bilevel optimization problem. The outer level objective is the selection of samples, and the inner objective is the optimization of model parameters. The representative works include Borsos et al. (2020), Glister (Killamsetty et al., 2021b), Zhou et al. (2022), and Retrieve (Killamsetty et al., 2021c). **Shapley value for Feature Selection:** These methods aim to select informative features and are widely used in Explainable AI (XAI). These methods (Cohen et al., 2005; Fryer et al., 2021) derive a metric from the weighted average over all contributions by the features but their effectiveness of coreset selection tasks has not been explored.

These methods perform well in some scenarios (specific subset fraction, specific outlier distribution, etc.) of full-precision training. However, none are verified on quantization-aware training settings and consider the requirements for QAT. We will further show that most of these methods cannot outperform random sampling in the QAT settings. Moreover, these methods either require early training of the target model for several epochs (Toneva et al., 2019; Paul et al., 2021) or time-consuming search (Sener & Savarese, 2018) and optimization (Killamsetty et al., 2021b;c), which leads to heavy computation overhead during QAT.

## 3 IMPORTANCE OF EACH SAMPLE IN QAT

In this section, we will first introduce quantization-aware training (QAT) and derive the gradient of real-value weights under cross-entropy and SGD in Sec. 3.1. We then analyze the change of gradient when a specific sample is removed from a training batch to investigate the importance of each training sample in Sec. 3.2. We propose an approximation of this gradient change considering the prediction error without introducing any memory overhead. We further prove that less training data is required when knowledge distillation is applied to QAT in Sec. 3.3. Another metric of importance based on the prediction gap between the quantized student and full-precision teacher model is introduced.

### 3.1 PRELIMINARIES OF QAT

During QAT, the real-value data $x^r$ is converted to $b$-bit quantized representation $x^q = q_b(x^r)$ by the quantizer $q_b$. Given the scale factor $s$ of the quantizer, the number of positive quantization levels $Q_P$,

and the number of negative quantization levels $Q_N$, we can have the quantizer $q_b$ as

$$x^q = q_b(x^r) = s \times \lfloor \text{clip}(x^r/s, -Q_N, Q_P) \rceil, \tag{1}$$

where $\lfloor \cdot \rceil$ is the rounding function that rounds the input to the nearest integer, $\text{clip}(x, r_{\text{low}}, r_{\text{high}})$ return $x$ with all value below $r_{\text{low}}$ set to be $r_{\text{low}}$ and all values above $r_{\text{high}}$ set to be $r_{\text{high}}$. For the unsigned quantization, $Q_N = 0, Q_P = 2^b - 1$. While for the quantization of signed data, $Q_N = 2^{b-1}, Q_P = 2^{b-1} - 1$. To solve the problem that the gradient cannot back-propagate in Equation 1 during QAT, the straight-through estimator (STE) (Bengio et al., 2013) is utilized to approximate the gradient. In the back-propagation of QAT with STE, the gradient of the loss $\mathcal{L}$ with respect to the real-value data $x^r$ is set to be

$$\frac{\partial \mathcal{L}}{\partial x^r} = \frac{\partial \mathcal{L}}{\partial x^q} \cdot \mathbf{1}_{-Q_N \leq x^r/s \leq Q_P}, \tag{2}$$

where $\mathbf{1}$ is the indicator function that outputs 1 within the quantization limit and 0 otherwise.

The training set of QAT is denoted as $\mathcal{T} = \{(x_i, y_i)\}_{i=1}^N$, where input $x_i \in \mathbb{R}^d$ are vectors of length $d$ and $y \in \{0, 1\}^M$ are one-hot vectors encoding labels. The neural network to be trained is denoted as $f(w^r, x)$, where real-value weights are $w^r$. We use cross-entropy loss $\mathcal{L}(\hat{p}, y) = \sum_{m=1}^M y^{(m)} \log p^{(m)}$ in the QAT and $p(w^q, x)$ is a probability vector denoting the output of the quantized neural network $f(w^q, x)$. Stochastic gradient descent (SGD) is used for the optimization. Suppose the real-value weights at iteration $t$ are $w_t^r$ and the batch of input samples is $\mathcal{T}_{t-1} \subseteq \mathcal{T}$, the weights are updated following

$$w_t^r = w_{t-1}^r - \eta \sum_{(x,y) \in \mathcal{T}_{t-1}} g_{t-1}(x, y), \tag{3}$$

where $\eta$ denotes the learning rate and $g_{t-1}(x, y) = \nabla \mathcal{L}_{t-1}(p(w_{t-1}^r, x), y)$ is the gradient of cross entropy loss.

## 3.2 ERROR VECTOR SCORE

To look into the importance of each sample $\{(x_i, y_i)\}$, we measure the difference of the expected magnitude of the loss vector on the training set $\mathcal{T}$ and another training set which only removes one specific sample $\mathcal{T}' = \mathcal{T} \setminus \{(x_i, y_i)\}$. For simplicity, we approximate all the training dynamics in continuous time. Based on the chain rule and STE of quantization, we have the change of loss $\mathcal{L}$ at time $t$ on sample $(x, y)$ of batch $\mathcal{T}$ as

$$\left. \frac{d\mathcal{L}}{dt} \right|_{(x,y),\mathcal{T}_t} = g_t(x, y) \frac{dw_t^q}{dt} = g_t(x, y) \frac{dw_t^r}{dt}. \tag{4}$$

According to the discrete-time dynamic of real-value weights in Eq. 3, we have $\frac{dw_t^q}{dt} \approx w_t^r - w_{t-1}^r = -\eta \sum_{(x,y) \in \mathcal{T}_{t-1}} g_{t-1}(x, y)$. To measure the contribution of a specific sample $(x_i, y_i)$, we measure the change of loss with and without the sample. For a given data batch $\mathcal{T}$, if the sample $(x_i, y_i) \notin \mathcal{T}$, we can ignore the change in $\left. \frac{d\mathcal{L}}{dt} \right|_{(x,y),\mathcal{T}_t}$. For any sample $(x_j, y_j) \in \mathcal{T}, j \neq i$ in the same batch, the importance $\mathcal{I}(x_i, y_i)$ is measured as

$$\mathcal{I}(x_i, y_i) = \left\| \left. \frac{d\mathcal{L}}{dt} \right|_{(x_j,y_j),\mathcal{T}} - \left. \frac{d\mathcal{L}}{dt} \right|_{(x_j,y_j),\mathcal{T}'} \right\|. \tag{5}$$

According to the chain rule, we have

$$\left. \frac{d\mathcal{L}}{dt} \right|_{(x_j,y_j),\mathcal{T}} = \frac{d\mathcal{L}(p(w_t^q, x_j), y_j)}{dw_t^q} \frac{dw_t^q}{dw_t^r} \frac{dw_t^r}{dt} \tag{6}$$

According to the characteristics of STE in Eq. 2, $\frac{dw_t^q}{dw_t^r} = 1$ holds for all input within the clipping range. Following the updating rule of weights in Eq. 3, the gradient $\frac{dw_t^r}{dt} = -\eta \sum_{(x^*,y^*) \in \mathcal{T}} g_{t-1}(x^*, y^*)$. The only difference of the training set $\mathcal{T}$ and $\mathcal{T}'$ is the existence of sample $(x_i, y_i)$. Thus, we have

$$\mathcal{I}(x_i, y_i) = \left\| \frac{d\mathcal{L}}{dw_t^q} \left( \left. \frac{dw_t^r}{dt} \right|_{(x_j,y_j),\mathcal{T}} - \left. \frac{dw_t^r}{dt} \right|_{(x_j,y_j),\mathcal{T}'} \right) \right\| = \eta \left\| \frac{d\mathcal{L}}{dw_t^q} \cdot g_{t-1}(x_i, y_i) \right\|. \tag{7}$$

We use $\frac{d\mathcal{L}}{dw_t^q}$ as a simplification of $\frac{d\mathcal{L}(p(w_t^q, x_j), y_j)}{dw_t^q}$, which is only dependant on the current training sample $(x_j, y_j)$ but not dependant on the sample $(x_i, y_i)$ that is removed from batch $\mathcal{T}$. Since learning rate $\eta$ and the gradient of loss w.r.t. quantized weights $\frac{d\mathcal{L}}{dw_t^q}$ are sample-agnostic, the importance of the sample $(x_i, y_i)$ in this batch $\mathcal{T}$ is only related to the gradient norm of cross-entropy loss of this sample $||g_{t-1}(x_i, y_i)||$. The examples with a larger gradient norm expectation have more influence on the supervised training of other data, which means they are important for QAT and should be included in the coreset. We can select data with high importance by sorting by the norm of gradient, which is also covered in previous works Paul et al. (2021). However, storing the loss gradient for comparison requires extra memory and is hard to transfer between network architectures. We approximate the norm gradient with the norm of the error vector, which is defined as follows.

**Definition 1 (Error Vector Score)** *The error vector score of a training sample $(x, y)$ at iteration $t$ id defined to be $d_{EVS} = ||p(w_t^q, x) - y||_2$.*

For any input $x \in \mathbb{R}^d$, gradient norm $||g_t(x, y)||$ is a non-random score. We take its expectation over random minibatch sequence and random initialization to get the expected gradient norm $\mathbb{E}||g_t(x, y)||$, which can also be indicated as

$$\mathbb{E}\,||g_t(x, y)|| = \sum_{m=1}^{M} \frac{d\mathcal{L}_t(p(w_t^q, x), y)^T}{df_t^{(m)}} \frac{df_t^{(m)}}{dw_t^q}, \tag{8}$$

where $\frac{df_t^{(m)}}{dw_t^q}$ denotes the gradient of $m$-th logit on weights. Since we are using cross-entropy as the loss function, the gradient of loss $\mathcal{L}$ on the $m$-th logit output $f_t^{(m)}$ follows $\frac{d\mathcal{L}_t(p(w_t^q, x), y)^T}{df_t^{(m)}} = p(w_t^q, x)^{(m)} - y^{(m)}$. Previous works (Fort et al., 2020; Fort & Ganguli, 2019) empirically observe that $\frac{df_t^{(m)}}{dw_t^q}$ are similar across different logits $m \in M$ and training sample $(x, y)$. Thus, the gradient norm is positively correlated to the error vector score $d_{EVS}$ during important sample selection. Different from the previous method (Paul et al., 2021) also leveraging the error metrics, no early training is required, and we only use the current quantized model prediction $p(w_t^q, x)$ during QAT.

### 3.3 DISAGREEMENT SCORE AND KNOWLEDGE DISTILLATION

Intrinsically, a quantized classification network should learn an ideal similar mapping $f$ from input sample $x$ to the output logits $f(w, x)$ as a full-precision network, and the gap between the quantized prediction $p(w^q, x)$ of student model and real-value prediction $p_{\mathbf{T}}(w^r, x)$ of teacher model $\mathbf{T}$ needs to be minimized. Based on this insight, knowledge distillation (KD) is widely used during QAT with a full-precision model as the teacher, which can also be seen in previous works (Polino et al.; Huang et al., 2022; Mishra & Marr, 2018; Liu et al., 2023). The loss function is designed to enforce the similarity between the output of the full-precision teacher and the quantized student model as

$$\mathcal{L}_{KD} = -\frac{1}{N} \sum_{m}^{M} \sum_{i=1}^{N} p_{\mathbf{T}}^{(m)}(w^r, x_i) \log(p^{(m)}(w^q, x_i)) \tag{9}$$

where the KD loss is defined as the cross-entropy between the output distributions $p_{\mathbf{T}}$ of a full-precision teacher and a quantized student on the same input $x$ but different weights representation $w^r$ and $w^q$. $x_i$ is one of the input samples from the training set. $m$ and $N$ denote the classes and total training sample numbers, respectively.

Note that this process can be regarded as the distribution calibration for the student network, and one-hot label is not involved during QAT. Since the loss function used for knowledge distillation is still cross-entropy, and we still assume we use SGD for the optimization, most conclusions in Sec. 3.2 still hold by replacing the one-hot ground truth label $y$ with full-precision teacher's prediction $p_{\mathbf{T}}(w_t^r, x)$. Thus, we propose the disagreement score as follows.

**Definition 2 (Disagreement Score)** *The disagreement score of a training sample $(x, y)$ at iteration $t$ is defined to be $d_{DS} = ||p(w_t^q, x) - p_{\mathbf{T}}(w_t^r, x)||_2$.*

The core difference between error vector score $d_{EVS}$ and disagreement score $d_{DS}$ is the target label. While $d_{EVS}$ uses one-hot hard labels, the $d_{DS}$ uses the distilled soft labels. We empirically notice that

the data needed for the training is reduced when knowledge distillation is applied, which is helpful for our coreset selection with a small data fraction. We will further demonstrate the advantage in terms of training data requirements using the soft label based on Vapnik–Chervonenkis theory (Vapnik, 1999), which decomposes the classification error $R(f_s)$ of a classifier $f_s \in \mathcal{F}_s$ as

$$R(f_s) - R(f) \leq O\left(\frac{|\mathcal{F}_s|_C}{n^{\alpha_s}}\right) + \varepsilon_s, \tag{10}$$

where $O(\cdot)$ denotes the asymptotic approximation and $\varepsilon_s$ is the approximation error of $\mathcal{F}_s$ with respect to $\mathcal{F}$. $f \in \mathcal{F}$ denotes the real target function. $|\cdot|_C$ is the VC-Dimension of the function class measuring its capacity. $n$ is the number of total training data. $\frac{1}{2} \leq \alpha_s \leq 1$ is an indicator measuring the difficulty of the problems. For non-separable and difficult problems, $\alpha_s = \frac{1}{2}$, which means the classifier learns at a slow rate of $O(n^{-\frac{1}{2}})$. For separable and easy problems, $\alpha_s = 1$, indicating the classifier learns at a fast rate. In our setting, if the quantized model $f_q \in \mathcal{F}_q$ directly learns from the hard labels, the difficulty of the problem is high, and we assume $\alpha_q = \frac{1}{2}$, we have

$$R(f_q) - R(f) \leq O\left(\frac{|\mathcal{F}_q|_C}{\sqrt{n}}\right) + \varepsilon_q, \tag{11}$$

where $\varepsilon_q$ is the approximation error of the quantized model. However, if we first train the full-precision teacher model $f_r \in \mathcal{F}_r$ and then utilize knowledge distillation to learn the representation from the teacher, the difficulty of the learning becomes easier, assuming $\alpha_r = 1$, we have

$$R(f_r) - R(f) \leq O\left(\frac{|\mathcal{F}_r|_C}{n}\right) + \varepsilon_r, R(f_q) - R(f_r) \leq O\left(\frac{|\mathcal{F}_q|_C}{n^{\alpha_{qr}}}\right) + \varepsilon_{qr}, \tag{12}$$

where the $\varepsilon_r, \varepsilon_{qr}$ denotes the approximation error of the $\mathcal{F}_r$ with respect to $\mathcal{F}$ and approximation error of the $\mathcal{F}_q$ with respect to $\mathcal{F}_r$, respectively. Compared to the direct learning quantized model from the hard label shown in Eq. 11, the knowledge distillation with real-value teacher $f_r$ yields the classification error as follows:

$$R(f_q) - R(f) \leq O\left(\frac{|\mathcal{F}_r|_C}{n}\right) + \varepsilon_r + O\left(\frac{|\mathcal{F}_q|_C}{n^{\alpha_{qr}}}\right) + \varepsilon_{qr} \leq O\left(\frac{|\mathcal{F}_q|_C + |\mathcal{F}_r|_C}{n^{\alpha_{qr}}}\right) + \varepsilon_r + \varepsilon_{qr}. \tag{13}$$

Following previous studies on knowledge distillation (Lopez-Paz et al., 2015; Mirzadeh et al., 2020), the soft labels contain more information than hard labels for each sample. Thus, we have $\varepsilon_r + \varepsilon_{qr} \leq \varepsilon_q$ and $O\left(\frac{|\mathcal{F}_q|_C + |\mathcal{F}_r|_C}{n^{\alpha_{qr}}}\right) \leq O\left(\frac{|\mathcal{F}_q|_C}{\sqrt{n}}\right)$. Combining these two inequalities, we have the inequality

$$O\left(\frac{|\mathcal{F}_q|_C + |\mathcal{F}_r|_C}{n^{\alpha_{qr}}}\right) + \varepsilon_r + \varepsilon_{qr} \leq O\left(\frac{|\mathcal{F}_q|_C}{\sqrt{n}}\right) + \varepsilon_q, \tag{14}$$

which means when the number of training samples $n$ is the same, the upper bound of classification error based on the soft label is lower. When we want to achieve the same upper bound of classification error $R(f_q) - R(f)$ using these two techniques, learning from soft labels requires less data. This is the core reason why we use knowledge distillation and disagreement score $d_{DS}$ to select the coreset.

## 4 ADAPTIVE CORESET SELECTION FOR QAT

In Sec. 3, we propose to use $d_{EVS}$ and $d_{DS}$ to select the coreset for QAT. While $d_{DS}$ could help select those samples that produce large performance gaps between quantized and full-precision models, $d_{EVS}$ targets more at the error of quantized prediction. These two metrics cover different characteristics of training data, and we need both to improve the diversity of our coreset. For different stages of QAT, the different metrics should be considered to select samples that fit the current training objective. Previous research (Kim et al., 2019) has shown that QAT should start with the hard label to help a better initialization for the quantized model and then use soft labels to guide it to better local minima. In light of this, we propose Adaptive Coreset Selection (ACS) for QAT to select the important samples considering current training epoch t, $d_{EVS}$, and $d_{DS}$ adaptively.

For the given current training epoch $t$ and the total training epoch $E$, we propose a cosine annealing weights coefficient $\beta(t) = \cos(\frac{t}{2E}\pi)$ to consider two metrics simultaneously and balance between them. The final selection metric is a linear combination of $d_{EVS}$ and $d_{DS}$ as follows:

$$d_{ACS}(t) = \beta(t)d_{EVS}(t) + (1 - \beta(t))d_{DS}(t). \tag{15}$$

As we have $\beta(0) = 1$ and $\beta(E) = 0$, in the early stage, the selection is mainly based on the error vector score $d_{\text{EVS}}$. When the quantized model converges, we focus more on the disagreement score $d_{\text{DS}}$ in later epochs. We perform coreset selection every $R$ epoch, where $R$ is determined before the training. The pseudo-code for our ACS algorithm is shown in Alg. 1.

There are two major advantages of our proposed quantization-aware ACS. The first lies in the adaptation to the training phase when knowledge distillation is applied. As soft labels retain more information about the target than hard labels, we should encourage the quantized student model to learn sequentially

---

**Algorithm 1** Adaptive Coreset Selection for QAT

**Input:** Training dataset $T = \{(x_i, y_i)\}_{i=1}^n$, Real-value network with weights $\mathbf{W^r}$, Coreset data fraction per epoch $S$, Total training epochs $E$, Selection interval $R$, Initial coreset $T_{\text{ACS}}(t) = \emptyset$
**Output:** Quantized network with weights $\mathbf{W^q}$
Initialize quantized weights $\mathbf{W^q}$ from $\mathbf{W^r}$ following Eq. 1
**for** $t \in [0, ..., E-1]$ **do**
  **if** $t\%R == 0$ **then**
    $\beta(t) = \cos(\frac{t}{2E}\pi)$
    **for** $(x_i, y_i) \in T$ **do**
      $d_{\text{EVS}}(x_i, t) = \|p(\mathbf{W_t^q}, \mathbf{x_i}) - \mathbf{y_i}\|_2$
      $d_{\text{DS}}(x_i, t) = \|p(\mathbf{W_t^q}, \mathbf{x_i}) - \mathbf{p_T}(\mathbf{W_t^r}, \mathbf{x_i})\|_2$
      $d_{\text{ACS}}(x_i, t) = \beta(t)d_{\text{EVS}}(x_i, t) + (1 - \beta(t))d_{\text{DS}}(x_i, t)$
    **end for**
    Sort $d_{\text{ACS}}(x_i, t)$, select top $S\%$ samples to replace $T_{\text{ACS}}(t)$
  **else**
    $T_{\text{ACS}}(t) \leftarrow T_{\text{ACS}}(t-1)$
  **end if**
  Train $\mathbf{W^q}$ on $T_{\text{ACS}}(t)$ following Eq. 9
**end for**

---

on hard labels first and soft labels then. This implicit learning hierarchy is observed in QKD (Kim et al., 2019) and is named "self-learning" and "tutoring". With the proposed ACS fully aware of this hierarchy, the selected coreset helps stabilize the training and guarantee faster convergence. The second advantage is related to the diversity of training data. More training samples could be covered in the coreset of different epochs, and the coverage of the original full dataset contributes to the development of a more robust model. Note that only when the optimal data sequence and high training sample diversity are achieved simultaneously the performance of QAT will be significantly better. We will demonstrate in the appendix that even when all data are covered but the order is random, the accuracy of our quantized model will be negatively influenced.

## 5 EXPERIMENTS

**Datasets and networks** The experiments are conducted on CIFAR-100 dataset (Krizhevsky et al., 2009) and ImageNet-1K dataset (Deng et al., 2009). We only perform basic data augmentation in PyTorch (Paszke et al., 2019), which includes *RandomResizedCrop* and *RandomHorizontalFlip* during training, and single-crop operation during evaluation. We evaluate MobileNetV2 (Howard et al., 2017) on the CIFAR-100 dataset and evaluate ResNet-18 (He et al., 2016) on the ImageNet-1K dataset. The *width multiplier* is set to be 0.5 for MobileNetV2.

**Baselines** For a fair comparison, we choose multiple coreset selection methods from different categories as our baseline. The selected methods include: Random Sampling, EL2N-Score (Paul et al., 2021), Forgetting (Toneva et al., 2019), Glister (Killamsetty et al., 2021b), kCenterGreedy (Sener & Savarese, 2018), Contextual Diversity (CD) (Agarwal et al., 2020), Moderate Coreset (Xia et al., 2023). For methods that involve early training, we set training epochs as 5. We give a more detailed introduction to the technical details of these methods in the appendix. We would like to emphasize that comparing adaptive-based and non-adaptive-based methods is fair and the common practice of previous research (Pooladzandi et al., 2022; Killamsetty et al., 2021c).

**Training and selection details** For MobileNetV2 on CIFAR-100, we train the network for 200 epochs using a learning rate of 0.01, weight decay of 5e-4, batch size of 512, and SGD optimizer. For ResNet-18 on ImageNet-1K, we train the network for 120 epochs using a learning rate of 1.25e-3, no weight decay, batch size of 512, and Adam optimizer. We use the quantization method and full-precision model following LSQ+ (Bhalgat et al., 2020). All the experiments were carried out on 2 NVIDIA RTX 3090 GPUs. For CIFAR-100 experiments, we choose $R = 20$, and each experiment of different settings is repeated 5 times. For ImageNet-1K experiments, we choose $R = 10$. We use knowledge distillation with the corresponding full-precision model as the teacher in all experiments, regardless of the method and dataset fraction.

### 5.1 BENCHMARKING PREVIOUS CORESET METHOD

The comparison of the QAT Top-1 accuracy of MobileNetV2 on CIFAR-100 and ResNet-18 on ImageNet-1K is shown in Tab. 1 and Tab. 2. We note that most previous methods cannot exceed random selection in our QAT setting, and the few surpass the baseline only on specific data fractions.

For example, kCenterGreedy (Sener & Savarese, 2018) shows satisfying performance when the subset size is large (50% for CIFAR-10, 70%/80% for ImageNet-1K) but fails to demonstrate effectiveness on small coreset size. Our method outperforms state-of-the-art methods on all subset fractions $\mathcal{S}$ by a great margin. Specifically, the ResNet-18 accuracy of 10% subset fraction on ImageNet-1K using our method is 68.39%, achieving an absolute gain of 4.24% compared to the baseline method. While our method with coreset cannot outperform full-data training ($\mathcal{S} = 100\%$), we will show in the appendix that we can achieve lossless acceleration on the dataset with noise.

Table 1: Comparison of Top-1 Accuracy of different coreset selection methods on QAT of quantized MobileNetV2 on CIFAR-100 dataset with different subset fraction. The bit-width is 2/32 for weights/activations. Each experiment is repeated 5 times with random seeds. The mean accuracy and standard deviation are reported. When full data are selected ($\mathcal{S} = 100\%$), the accuracy is 68.07±0.9.

| Method/Fraction (%) | 10% | 20% | 30% | 40% | 50% |
|---|---|---|---|---|---|
| Random | 62.25±0.91 | 64.07±0.65 | 65.22±0.33 | 65.55±0.64 | 66.24±0.94 |
| EL2N Score (Paul et al., 2021) | 63.02±0.62 | 64.04±1.56 | 65.56±0.75 | 64.55±0.81 | 65.31±0.37 |
| Forgetting (Toneva et al., 2019) | 60.71±0.33 | 63.07±0.59 | 65.04±0.80 | 65.26±0.84 | 65.96±1.24 |
| Glister (Killamsetty et al., 2021b) | 56.54±1.16 | 60.37±1.05 | 61.32±0.55 | 63.03±0.98 | 64.78±0.91 |
| kCenterGreedy (Sener & Savarese, 2018) | 60.15±0.83 | 62.65±0.51 | 63.78±0.65 | 64.83±0.59 | 66.27±1.10 |
| CD (Agarwal et al., 2020) | 60.33±0.78 | 62.62±0.90 | 64.04±1.03 | 64.78±0.38 | 65.31±0.37 |
| Moderate (Xia et al., 2023) | 57.92±0.27 | 60.43±0.66 | 62.83±0.61 | 63.56±0.75 | 64.45±1.55 |
| Ours | **63.67±0.77** | **65.91±0.72** | **66.41±0.75** | **66.85±0.49** | **67.19±0.54** |

Table 2: Comparison of Top-1 Accuracy of different coreset selection methods on QAT of quantized ResNet-18 on ImageNet-1K with different subset fraction. The bitwidth for quantized ResNet-18 is 4/4 for weights/activations. When full data are selected ($\mathcal{S} = 100\%$), the accuracy is 72.46.

| Method/Fraction (%) | 10% | 30% | 50% | 60% | 70% | 80% |
|---|---|---|---|---|---|---|
| Random | 64.15 | 68.53 | 70.49 | 70.94 | 71.06 | 71.96 |
| EL2N Score (Paul et al., 2021) | 61.71 | 67.31 | 70.14 | 70.89 | 71.54 | 71.88 |
| Forgetting (Toneva et al., 2019) | 63.09 | 67.77 | 70.14 | 71.00 | 71.36 | 71.82 |
| Glister (Killamsetty et al., 2021b) | 63.25 | 68.94 | 70.92 | 71.39 | 71.93 | 72.22 |
| kCenterGreedy (Sener & Savarese, 2018) | 62.98 | 68.56 | 70.35 | 71.13 | 71.59 | 71.96 |
| CD (Agarwal et al., 2020) | 63.22 | 68.74 | 70.90 | 71.27 | 71.78 | 72.11 |
| Moderate (Xia et al., 2023) | 62.39 | 68.06 | 70.43 | 70.62 | 71.56 | 71.99 |
| Ours | **68.39** | **71.09** | **71.59** | **72.00** | **72.19** | **72.31** |

**Efficiency analysis and effect of adaptive epochs** The choice of selection interval $R$ is vital in our algorithm, as too large $R$ will fail to help adaption and improve data diversity, and too small $R$ will introduce too much computation overhead and undermine the efficiency. We apply grid search on $R$ and empirically prove that $R = 10$ is

Table 3: Analysis on adaptive epochs $R$.

| Fraction | $\mathcal{S}$=10% | | $\mathcal{S}$=30% | | $\mathcal{S}$=50% | | $\mathcal{S}$=70% | |
|---|---|---|---|---|---|---|---|---|
| | Acc. | Time | Acc. | Time | Acc. | Time | Acc. | Time |
| $R = 5$ | 68.8 | 12.7h | 71.05 | 21.5h | 71.47 | 37.0h | 72.15 | 49.1h |
| $R = 10$ | 68.39 | 11.3h | 71.09 | 20.7h | 71.59 | 36.1h | 72.19 | 48.0h |
| $R = 20$ | 67.58 | 11.0h | 70.35 | 20.2h | 71.42 | 35.5h | 71.97 | 47.6h |
| $R = 40$ | 66.10 | 10.7h | 69.86 | 19.8h | 71.25 | 34.9h | 72.00 | 47.2h |
| $R = 60$ | 64.96 | 10.5h | 69.27 | 19.5h | 71.05 | 34.4h | 71.93 | 46.9h |
| $R > 120$ | 62.82 | 10.3h | 67.62 | 19.2h | 69.98 | 33.9h | 71.37 | 46.6h |

optimal for our ImageNet-1K training. The accuracy and training time results are shown in Tab. 3. We can observe from the results that $R = 10$ achieves a similar performance as $R = 5$ with a shorter training time. As no back-propagation of the gradient is involved during the computation of $d_{\text{EVS}}$ and $d_{\text{DS}}$, the computation overhead is acceptable under most settings with $R > 5$ compared with previous methods involving training. The only method from the selected baselines that has similar efficiency to ours is Moderate Coreset (Xia et al., 2023), which also only forwards on samples and sorts the distance metrics. We report the detailed training time of all previous methods in the appendix.

**Ablation study and effect of annealing strategy** As we propose two metrics for coreset selection: $d_{\text{EVS}}$ and $d_{\text{DS}}$. Therefore, it is important to analyze how we should balance between them and the contribution of both metrics. We try the following different annealing strategies and settings: (1) fixed ($\beta(t) = 0.5$); (2) linear ($\beta(t) = 1 - \frac{t}{E}$); (3) square root ($\beta(t) = 1 - \sqrt{\frac{t}{E}}$); (4) quadratic ($\beta(t) = 1 - (\frac{t}{E})^2$); (5) cosine ($\beta(t) = \cos(\frac{t}{2E}\pi)$); (6) $d_{\text{EVS}}$ only ($\beta(t) = 1$); (7) $d_{\text{DS}}$ only ($\beta(t) = 0$); The results of coreset selection of 4-bit quantized ResNet-18 on ImageNet-1K are listed in Tab. 4. The performance gap

Table 4: Analysis on strategy $\beta(t)$.

| Fraction (%) | 10% | 30% | 50% | 70% |
|---|---|---|---|---|
| fixed | 67.95 | 70.83 | 71.21 | 72.01 |
| linear | 68.37 | **71.11** | 71.46 | 72.18 |
| sqrt | 68.15 | 71.03 | 71.44 | 72.15 |
| quadratic | 68.11 | 70.95 | 71.35 | 72.10 |
| $d_{\text{EVS}}$ only | 68.06 | 70.63 | 71.41 | 71.96 |
| $d_{\text{DS}}$ only | 67.07 | 70.24 | 71.43 | 72.08 |
| cosine | **68.39** | 71.09 | **71.59** | **72.19** |

with different annealing strategies is close as they all follow the trends to use $d_{EVS}$ in the early epochs of training and $d_{DS}$ in the late training epochs. Among all these annealing strategies, cosine annealing is slightly better. When only one metric is used for the selection, the performance will drop. We also notice that $d_{EVS}$ and $d_{DS}$ are complementary as $d_{EVS}$ works well with small data fraction and $d_{DS}$ has better performance with large data fraction.

**Visualization and Analysis**   We visualize the loss landscape (Li et al., 2018) of MobileNetV2 training on the full CIFAR-100 dataset, 10% random subset of CIFAR-100, and 10% coreset of CIFAR-100 based on our methods shown in Fig. 2. We can see from the results that QAT on coreset with our method has a more centralized and smoother loss compared to the baseline methods, which reflects that our method helps improve the training stability of QAT.

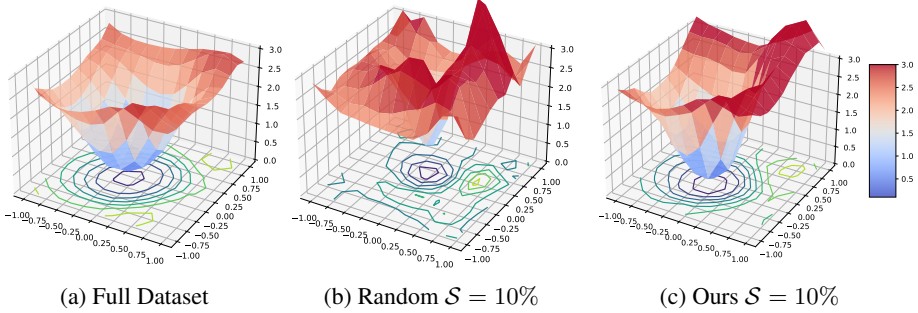

(a) Full Dataset          (b) Random $\mathcal{S} = 10\%$          (c) Ours $\mathcal{S} = 10\%$

Figure 2: Visualization of the loss landscape (Li et al., 2018) of 2-bit quantized MobileNetV2 trained on the CIFAR-100 (a) full dataset, (b) random subset, and (c) our coreset.

The Top-1 Accuracy comparison of different methods, networks, and datasets is visualized in Fig. 3a and Fig. 3b, which is a straightforward demonstration of the remarkable selection quality of our method ACS. We also visualize the distribution of disagreement scores $d_{DS}$ and error vector score $d_{EVS}$ in Fig. 3c and Fig. 3d. The setting is the same as the MobileNetV2 experiment listed in Tab. 1. We can see from the results that the mean of $d_{DS}$ shifts to zero during the QAT, which proves that $d_{DS}$ is a useful metric to quantify the importance of each sample. The distribution discrepancy between $d_{EVS}$ and $d_{DS}$ proves the necessity of considering both metrics to select diverse data into our coreset.

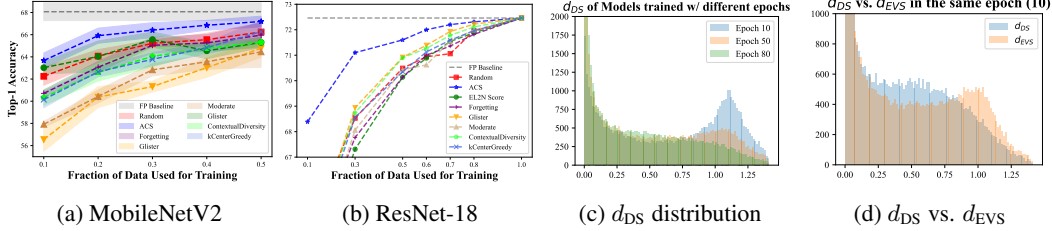

(a) MobileNetV2          (b) ResNet-18          (c) $d_{DS}$ distribution          (d) $d_{DS}$ vs. $d_{EVS}$

Figure 3: (a) The Top-1 Accuracy of the 2/32(W/A)-bit quantized MobileNetV2 on the CIFAR-100 with different data fractions. (b) The Top-1 Accuracy of the 4/4(W/A)-bit quantized ResNet-18 on the ImageNet-1K with different data fractions. (c) Disagreement scores $d_{DS}$ Distribution of MobileNetV2 for different epochs (epochs 10, 50, and 80). (d) Comparison of the distribution of disagreement scores $d_{DS}$ and error vector score $d_{EVS}$ of MobileNetV2 in the same epoch (epoch 10).

## 6 CONCLUSION

This is the first work focusing on the training and data efficiency of quantization-aware training (QAT) from the perspective of coreset selection. By removing samples from the training batches and analyzing the loss gradient, we theoretically prove that the importance of each sample varies significantly for QAT. Error-vector score and disagreement score are proposed to quantify this importance. Considering the training characteristics of QAT, we propose a fully quantization-aware Adaptive Coreset Selection (ACS) method to better adapt to different training phases and improve training data diversity. Extensive experiments on various datasets, networks, and quantization settings further demonstrate the effectiveness of our method.

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

## APPENDIX

This appendix includes an additional introduction on related works, training dynamics, efficiency analysis, extended experimental analysis on data coverage, and a discussion of limitations and broader impacts not included in the main text due to space limitations. These contents are organized in separate sections as follows:

- Sec. A elaborates the detail of previous representative coreset selection methods and analyzes the reason for failure in quantization-aware training (QAT) scenario.
- Sec. B demonstrate the training dynamics of our ACS and show how our methods help to stabilize the training and accelerate the convergence.
- Sec. C discuss why our method cannot achieve lossless acceleration compared to full-data training **in ideal cases** based on the neural scaling law. The experiments on the dataset with label noise are provided to prove the superiority of our method **in real-world scenarios**.
- Sec. D includes the real training time of QAT and the training time composition (selection vs. training) on NVIDIA RTX 3090 GPUs with our method and all baseline methods to prove that our approach is also optimal in efficiency.
- Sec. E provides additional experiments to prove that the improvement of performance with our method does not exclusively come from covering more data from the full training set.
- Sec. F provides detailed experimental results with and without knowledge distillation (KD). Both the accuracy and real training time are shown.
- Sec. G analyzes the transferability and generalizability of our coreset from different models.
- Sec. H shows the effectiveness of our method on additional object detection tasks of RestinaNet on the COCO object detection benchmark.
- Sec. I discusses our method's limitations and broader impacts.

## A   DETAILED INTRODUCTION OF SELECTED BASELINES

In this section, we will introduce the selected baseline coreset selection methods in detail and show the defects of these methods when applied to QAT.

**EL2N-Score/GraNd-Score (Paul et al., 2021)**   The GraNd-score for a given training sample $\{x, y\}$ is defined as $\chi_t(x, y) = E_{w_t} \|g_t(x, y)\|_2$, which is the expected magnitude of the loss vector with respect to the weights. The importance of each sample is measured by the expected loss gradient norm, which has a similar intuition to our error-vector score $d_{\text{EVS}}$. However, another assumption from GraNd score is that this approximation only holds when the model has been trained for a few epochs. We must perform early training on the current model to have statistics to compute the score. In addition, storing all the gradients incurs significant memory overheads during QAT. The efficiency will be lower than full dataset training with a high subset fraction. The performance with these metrics is sub-optimal as the converged quantized model in the later training epochs of QAT is not considered.

**Forgetting (Toneva et al., 2019)**   The core contribution of Forgetting is that forgetting and learning events are defined. In the classification setting with a given dataset $\mathcal{D} = (\mathbf{x}_i, y_i)_i$. For training example $\mathbf{x}_i$ obtained after $t$ steps using SGD, the predicted label is denoted as $\hat{y}_i^t = \arg\max_k p(y_{ik}|\mathbf{x}_i; \theta^t)$. $\text{acc}_i^t = 1_{\hat{y}_i^t = y_i}$ is defined to be a binary indicator encoding whether the classification of the specific example is correct at time step $t$. Example $\mathbf{x}_i$ undergoes a *forgetting event* when $\text{acc}_i^t$ decreases between two consecutive updates: $\text{acc}_i^t > \text{acc}_i^{t+1}$. The example $\mathbf{x}_i$ is misclassified at step $t + 1$ after having been correctly classified at step $t$. Corresponding to the forgetting event, a *learning event* is defined to be $\text{acc}_i^t < \text{acc}_i^{t+1}$. With the statistics of forgetting events in early training, we can select those samples that incur more forgetting events and are more difficult to learn. However, the intuition to select "difficult" samples that incur more misclassification is not always correct. For large-scale datasets (ImageNet-1K, etc.) and datasets with a relatively smaller size (such as MNIST, CIFAR-10, etc.). the difficulties of classification vary significantly. Selecting samples that are difficult to learn at the very beginning is not always reasonable. In the quantization-aware training setting, the quantized

model will first calibrate the quantization parameters and recover the weight at the early stages. We should not select "difficult samples" for calibration and recovery.

**Glister (Killamsetty et al., 2021b)**  The Glister performs data selection based on the assumption that the inner discrete data selection is an instance of (weakly) submodular optimization. Let $V = \{1, 2, \cdots, n\}$ denote a ground set of items (the set of training samples in the setting of coreset selection). Set functions are defined as functions $f : 2^V \to \mathbf{R}$ that operate on subsets of $V$. A set function $f$ is defined to be a submodular function (Fujishige, 2005) if it satisfies the diminishing returns property that for subsets $S \subseteq T \subseteq V, f(j|S) \triangleq f(S \cup j) - f(S) \geq f(j|T)$. Some natural combinatorial functions (facility location, set cover, concave over modularity, etc.) are submodular functions (Iyer et al., 2020). Submodularity is also very appealing because a simple greedy algorithm achieves a $1 - 1/e$ constant factor approximation guarantee (Nemhauser et al., 1978) for the problem of maximizing a submodular function subject to a cardinality constraint (which most data selection approaches involve). Moreover, several variants of the greedy algorithm have been proposed, further scaling up submodular maximization to almost linear time complexity (Minoux, 1978; Mirzasoleiman et al., 2013). However, the greedy algorithm still consumes tremendous time compared to other metric-based selection methods, which only need to perform sorting on the proposed metrics. We also observed that the greedy algorithm fails on QAT for low-bit settings.

**kCenterGreedy (Sener & Savarese, 2018)**  As one of the most straight-forward geometry-based coreset selection methods, the intuition is simple: we can measure the similarity by the distance of data points and select the center point of data cluster can make use of the redundancy in the dataset. This method aims to solve the *minimax facility location* problem (Wolf, 2011), which is defined to be selecting $k$ samples as subset $S$ from the full dataset $T$ such that the longest distance between a data point in $T \backslash S$ and its closest data point in $S$ is minimized:

$$\min_{S \subset T} \max_{x_i \in T \backslash S} \min_{x_j \in S} \mathcal{D}(x_i, x_j), \tag{16}$$

where $\mathcal{D}(\cdot, \cdot)$ is the distance measurement function. The problem is NP-hard, and a greedy approximation known as K-CENTER GREEDY has been proposed in Sener & Savarese (2018). Similarly, as the greedy algorithm is involved, the efficiency is negatively influenced, which is not affordable for our QAT setting.

**Contextual Diversity (CD) (Agarwal et al., 2020)**  Specifically designed for coreset selection for deep convolutional neural networks (CNNs), Contextual Diversity (CD) fully leverages the ambiguity in feature representations. The ambiguity is quantified as class-specific confusion as the selection metric. Assume $C = \{1, \ldots, n_C\}$ is the set of classes predicted by a CNN-based model. For a region $r$ within an input image $I$, let $P_r = P_r(\widehat{y} \mid I; \theta)$ be the softmax probability vector as predicted by the model $\theta$. The pseudo-label for the region $r \subseteq I$ is defined as $\widehat{y}_r = \arg \max_{j \in C} P_r[j]$, where the notation $P_r[j]$ denotes the $j^{th}$ element of the vector. For a given model $\theta$ over the unlabeled $I$, the class-specific confusion for class $c$ is defined as $P_I^c = \frac{1}{|I^c|} \sum_{I \in I^c} \left[ \frac{\sum_{r \in R_I^c} w_r P_r(\widehat{y}|I;\theta)}{\sum_{r \in R_I^c} w_r} \right]$ with $w_r \geq 0$ as the mixing weights. The pairwise contextual diversity, which is the KL-divergence of the metric between two samples $I_1$ and $I_2$ could be used as a distance metric to replace the Euclidean distance in the previous kCenterGreedy (Sener & Savarese, 2018). As this work basically follows kCenterGreedy (Sener & Savarese, 2018) to perform coreset selection, the defects also lie in the efficiency of the greedy algorithm and failure with the low-bit setting.

**Moderate Coreset (Xia et al., 2023)**  As the most recent work of coreset selection, Moderate has optimal efficiency as no greedy searching or back-propagation is involved during selection. Previous methods rely on score criterion that only applies to a specific scenario. This work proposes to utilize the score median as a proxy of the statistical score distribution and select the data points with scores close to the score median into the coreset. The main drawback of this method applied to quantization-aware training is that quantized distribution is not considered.

# B   QAT Training Dynamics with ACS

In this section, we report the training dynamics, including the training loss and training accuracy of QAT ResNet-18 on ImageNet-1K coreset with a 10% subset fraction. As can be seen from the results in Figure. 4, when the coreset changes adaptively to the current iterations, the accuracy will drop, and loss will increase significantly at the specific iteration of updating the subset. However, the quantized model will converge fast on the new subset. The training with an adaptive coreset can effectively help avoid overfitting and improve the final performance.

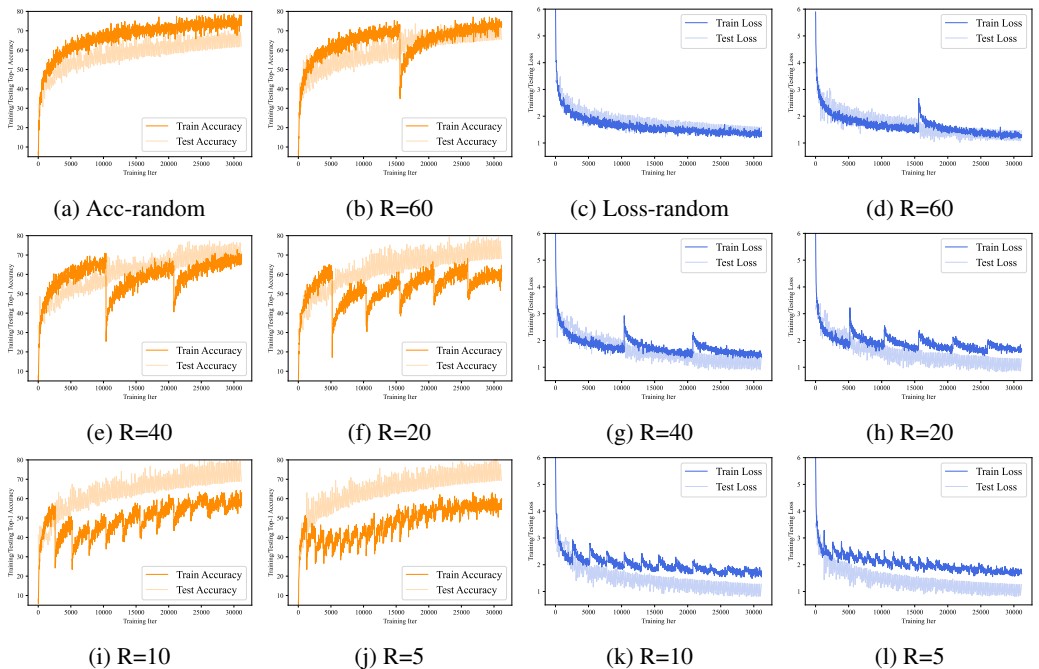

Figure 4: Training/Testing loss/accuracy comparison of 4/4-bit quantized ResNet-18 on 10% coreset of ImageNet-1K with different selection interval R.

# C   Discussion on lossless acceleration with coreset selection

In the results of Top-1 accuracy on CIFAR-100 and ImageNet-1K dataset shown in Table. 1 and Table. 2, our method significantly outperform all previous SOTA coreset methods. However, we still cannot achieve lossless acceleration compared to full-data training. We would like to clarify that all existing coreset selection methods cannot achieve lossless acceleration when the model size and training settings (training epochs, hyper-parameters, etc.) are appropriate.

**In an ideal case,** the real advantage of coreset selection is to achieve an optimal trade-off between model performance and training efficiency. Previous coreset research tries to minimize the accuracy decrease when training efficiency is improved. A theory for this accuracy-efficiency trade-off is the neural scaling law Kaplan et al. (2020); Alabdulmohsin et al. (2022), where the test loss $L(D)$ of a model follows a power law to the dataset size $D$ as $L(D) = (D_c/D)^\alpha$. The $D_c$ is a constant and $\alpha \in (0, 1)$ reflects that when the training data amount $D$ is reduced, the test loss $L(D)$ will always increase. What we do in this work is try to minimize $\alpha$ so that we can always significantly improve efficiency with only negligible accuracy loss in QAT.

**In real-world scenarios**, as mentioned in Kaplan et al. (2020), when there is overfitting in the training, we can achieve lossless acceleration compared to full data training because the performance is not bounded by the data size, which is a highly possible case in QAT. In addition, we would like to highlight another application of our method: identifying noise labels. The nature of advantage of our method is to select important training examples for QAT and remove those low-quality redundant examples. This is especially useful when there is label noise in the training set (some examples are

not correctly labeled). This application is proposed in previous coreset research Mirzasoleiman et al. (2020), but not verified on QAT. We follow the setting of Mirzasoleiman et al. (2020) to experiment on the QAT of ResNet-18 on CIFAR-10 with 10% randomized labels (labels of 10% of training samples are randomly re-generated). The ResNet-18 is quantized to 2/32 for weights are activations. We train the quantized ResNet-18 for 200 epochs using a learning rate of 0.01, weight decay of 5e-4, batch size of 512, and SGD optimizer on one NVIDIA RTX 3090 GPU. The experiments are repeated 5 times with random seeds under different data fractions and the mean accuracy and standard deviation are reported. When full data are selected, the accuracy is 89.91±0.51%. The comparison of the QAT Top-1 accuracy (%) of 2/32-bit quantized ResNet-18 on CIFAR-10 with 10% randomized labels is shown as follows in Table. 5:

Table 5: Comparision of Top-1 Accuracy on CIFAR-10 with 10% random label noise

| Method/Fraction (%) | 10% | 20% | 30% | 40% | 50% |
|---|---|---|---|---|---|
| Random | 83.14±0.51 | 84.77±0.80 | 84.82±0.57 | 85.09±0.15 | 85.33±0.60 |
| EL2N Score | 85.80±0.51 | 86.02±0.55 | 87.18±0.35 | 87.51±0.29 | 88.01±0.45 |
| Ours | **88.31±0.27** | **89.53±0.62** | **89.95±0.39** | **90.21±0.39** | **90.23±0.18** |

As can be seen from the results in Table. 5, our method outperforms all other selection baselines and even performs better than full-data training when coreset size $\mathcal{S} \geq 30\%$. This shows that our method can successfully prune those samples with incorrect labels. In this case, we actually achieve a "lossless acceleration" as both the accuracy and efficiency are better than full-data training.

## D  DETAILED EFFICIENCY COMPARISION

The detailed QAT training time of ResNet-18 on ImageNet-1K coreset and 2 NVIDIA RTX 3090 GPUs with different methods is listed in Table. 6. The only method with comparable efficiency to ours is the Moderate Coreset (Xia et al., 2023), which only needs to perform forwarding on samples and can eliminate optimization or greedy searching. The results prove that our method can effectively reduce training time without incurring computation overhead by the selection algorithm.

Table 6: Comparision of training time of different coreset selection methods on QAT of quantized ResNet-18 on ImageNet-1K with different subset fractions. The bitwidth for quantized ResNet-18 is 4/4 for weights/activations. When full data are selected ($\mathcal{S} = 100\%$), the training time is 62.3h.

| Method/Fraction (%) | 10% | 30% | 50% | 60% | 70% | 80% |
|---|---|---|---|---|---|---|
| EL2N Score (Paul et al., 2021) | 12.7h | 23.8h | 39.1h | 44.1h | 49.9h | 56.0h |
| Forgetting (Toneva et al., 2019) | 12.7h | 23.9h | 39.5h | 44.7h | 50.2h | 56.6h |
| Glister (Killamsetty et al., 2021b) | 13.8h | 23.0h | 39.9h | 46.6h | 58.0h | 66.8h |
| kCenterGreedy (Sener & Savarese, 2018) | 13.0h | 25.2h | 38.4h | 43.8h | 50.5h | 56.9h |
| CD (Agarwal et al., 2020) | 11.9h | 24.5h | 37.1h | 42.5h | 48.9h | 55.0h |
| Moderate (Xia et al., 2023) | 11.2h | 22.3h | 36.2h | 42.5h | 48.2h | 53.7h |
| Ours (R=10) | 11.3h | 20.7h | 36.1h | 42.2h | 48.0h | 53.8h |

We also provide the detailed training time composition of training and selection in Table. 7. Since KD will change the loss function of SGD training and is difficult to break down these two parts, we provide two sets of results with and without KD. The experiments are also conducted on quantized ResNet-18 on ImageNet-1K coreset with 2 NVIDIA RTX 3090 GPUs.

The results show that coreset selection only incurs a minimal efficiency overhead both with and without KD. The selection time is a constant and is only related to the full dataset size and the selection intervals $R$. The training with KD and without KD only influence the backpropagation time of the training and there is no backward in the selection phase, which makes the time of selection the same (3.2h) across all settings.

Table 7: Composition of training time on QAT of quantized ResNet-18 on ImageNet-1K with different subset fractions and KD settings. The bitwidth for quantized ResNet-18 is 4/4 for weights/activations.

| Stage/Fraction (%) | Apply KD? | 10% | 30% | 50% | 60% | 70% | 80% |
|---|---|---|---|---|---|---|---|
| Ours-all | ✓ | 11.3h | 20.7h | 36.1h | 42.2h | 48.0h | 53.8h |
| Ours-selection | ✓ | 3.2h | 3.2h | 3.2h | 3.2h | 3.2h | 3.2h |
| Ours-training | ✓ | 8.1h | 17.5h | 32.9h | 39.0h | 44.8h | 50.6h |
| Ours-all | | 9.7h | 16.0h | 29.5h | 34.4h | 40.2h | 45.4h |
| Ours-selection | | 3.2h | 3.2h | 3.2h | 3.2h | 3.2h | 3.2h |
| Ours-training | | 6.5h | 15.8h | 26.3h | 31.2h | 37.0h | 42.2h |

## E  CORESET COVERAGE ANALYSIS

The advantages of the proposed quantization-aware Adaptive Coreset Selection (ACS) algorithms are two-fold: adaption and diversity. In this section, we further demonstrate that improving diversity by covering all the training data in different subsets is not optimal compared to our adaptive coreset. We use the "Full Coverage Split" of the CIFAR-100 dataset as our baseline, which is to select non-overlapped samples randomly into the subset of fraction $S$ in the first $\lceil 1/S \rceil$ epochs. It is guaranteed that all the training samples are included, but the sequence is random in this setting. We apply QAT to a 2/32-bit quantized MobileNet-V2 on the random subset, "Full Coverage Split" subset, and coreset using our methods. The selection interval $R$ is the same across all settings. When the subset fraction $S$ is large, the difference in coverage rate with various methods is minor. We then only evaluate on the $S \in \{10\%, 20\%, 30\%, 40\%, 50\%\}$ The results are shown in Table. 8, where we can see that "Full Coverage Split" has limited superiority on performance compared to random baseline, especially when the subset fraction R is large. Our method outperforms the other two settings across all fractions, proving that both adaption and diversity help to improve the performance.

Table 8: Comparision of Top-1 Accuracy with Random vs. Full coverage split vs. Ours

| Method/Fraction (%) | 10% | 20% | 30% | 40% | 50% |
|---|---|---|---|---|---|
| Random | 62.25±0.71 | 64.07±0.47 | 65.22±0.41 | 65.55±0.80 | 66.24±0.55 |
| Full Coverage Split | 62.55±0.65 | 64.22±0.81 | 65.34±1.17 | 65.69±0.69 | 66.20±0.89 |
| Ours | **63.37±0.55** | **65.91±0.40** | **66.41±0.31** | **66.86±0.72** | **67.19±0.65** |

## F  DETAILED EXPERIMENTAL RESULTS ITHOUT KNOWLEDGE DISTILLATION

Knowledge Distillation (KD) is a normal and "default" setting for all previous quantization work (Mishra & Marr, 2018; Zhuang et al., 2020; Liu et al., 2022; Bhalgat et al., 2020; Huang et al., 2022), including the LSQ quantization (Bhalgat et al., 2020) we use in this paper. For a fair comparison with previous work, we equally utilize KD for all the experiments in this work regardless of the precision and dataset fraction. The full-data training baseline also involves training with knowledge distillation.

To verify the effectiveness of our method without KD, we remove the knowledge distillation in our method. Since the DS is built on the soft label of the teacher model, we do not use it in our selection w/o KD. Only EVS is applied as the selection metric. We follow the same settings for quantized ResNet-18 on ImageNet-1K as shown in our paper. The training time and accuracy are shown as follows in Table. 9. When full data are selected ($S = 100\%$), the Top-1 accuracy is 70.21% and training time is 3.1h without KD. From the results shown in Table. 9, we can see that our method still outperforms previous SOTA without KD and the training efficiency is still optimal.

Table 9: Comparison of Top-1 Accuracy and the training time of QAT of quantized ResNet-18 on ImageNet-1K with different subset fractions without KD settings. The bitwidth for quantized ResNet-18 is 4/4 for weights/activations.

| MethodFraction (%) | 10% | | 30% | | 50% | | 60% | | 70% | | 80% | |
|---|---|---|---|---|---|---|---|---|---|---|---|---|
| | Acc | Time | Acc | Time | Acc | Time | Acc | Time | Acc | Time | Acc | Time |
| EL2N w/o KD | 59.71 | 10.6h | 63.50 | 17.1h | 65.19 | 31.3h | 66.38 | 35.0h | 67.90 | 42.9h | 69.01 | 48.0h |
| Glister w/o KD | 62.41 | 11.9h | 65.18 | 22.7h | 66.47 | 34.5h | 67.05 | 41.7h | 68.81 | 50.3h | 69.45 | 56.9h |
| Ours w/o KD | 66.91 | 9.7h | 68.77 | 16.0h | 69.25 | 29.5h | 69.66 | 34.4h | 69.85 | 40.2h | 70.03 | 45.4h |

## G   CORESET TRANSFERABILITY AND GENERALIZABILITY

The theoretical analysis of the importance of each sample in QAT is model-specific, which means that data selected using our method is the optimal coreset for the current model. However, if the coreset discovered by a specific pre-trained model is applicable to other models, our method can be a potential solution for the model-agnostic coreset method. In addition, if there is a significant coreset overlap across different models, our method can even solve dataset distillation.

We first design experiments to verify the transferability of the coreset using our method. We use 2/32-bit ResNet-18 to generate a coreset on CIFAR-100 and apply it to MobileNet-V2. The results are shown in Table. 10. As the ResNet-18 coreset performs better than the random subset on MobileNet-V2, our coreset has some extent of generalization ability to unseen network structures, but the effectiveness is still worse than the MobileNetV2 coreset.

Table 10: Comparision of Top-1 Accuracy with different coreset of MobileNet-V2 on CIFAR-100.

| Method | Coreset Generation Model | 10% | 20% | 30% | 40% | 50% |
|---|---|---|---|---|---|---|
| Random | - | 62.25±0.71 | 64.07±0.47 | 65.22±0.41 | 65.55±0.80 | 66.24±0.55 |
| Ours | MobileNetV2 | 63.37±0.55 | 65.91±0.40 | 66.41±0.31 | 66.86±0.72 | 67.19±0.65 |
| Ours | ResNet-18 | 62.94±0.45 | 64.18±0.73 | 65.70±0.40 | 65.90±0.51 | 66.86±0.42 |

We then design experiments to verify the overlap between the coreset of different models. We analyze the 10% fraction coreset selected using our method in the final epochs of 2/32-bit ResNet-18, ResNet-50, and MobileNet-V2. The percentage of the overlap data is shown in Table. 11.

Table 11: Coreset Overlap of different model pairs on CIFAR-100.

| Model Pair | Coreset Overlap Percentage |
|---|---|
| ResNet-18, MobileNetV2 | 37.1% |
| ResNet-18, ResNet-50 | **77.3%** |
| ResNet-50, MobileNetV2 | 29.0% |

We empirically find out that there is some overlap between the coreset of different models, and the overlap is more significant when these two models have a similar structure (ResNet-18 and ResNet-50). We then further apply majority voting from the CIFAR-100 coreset of five models (ResNet-18, ResNet-34, ResNet-50, MobileNet-V2, MobileNet-V3) to generate a "general coreset". This "general coreset" of 10% data fraction is applied to a 2/32-bit quantized Vision Transformer ViT-B/16 (Dosovitskiy et al., 2020), where we only got a Top-1 accuracy of 72.3%, which is lower than 10% coreset using our method of 78.9% and even lower than 10% random coreset of 74.6%. We conclude that our method is model-specific, which has some extent of generalization ability to unseen network structures but cannot be a general data distillation approach.

## H   ADDITIONAL EXPERIMENTAL RESULTS

We further conduct experiments of 4/4-bit QAT of RetinaNet (Lin et al., 2017) with ResNet-18 backbone on the MS COCO object detection dataset (Lin et al., 2014). The QAT method used is FQN (Li et al., 2019a). The results of mAP are listed in Table. 12. Our method outperforms the

random selection baseline and state-of-the-art selection method Moderate significantly, which proves that our method is effective on object detection tasks.

Table 12: Comparision of performance with different methods of RetinaNet on COCO benchmarks.

| Method | Fraction | AP | $AP^{0.5}$ | $AP^{0.75}$ | $AP^S$ | $AP^M$ | $AP^L$ |
|---|---|---|---|---|---|---|---|
| Full | 100% | 28.6 | 46.9 | 29.9 | 14.9 | 31.2 | 38.7 |
| Random | 50% | 21.4 | 39.8 | 20.4 | 7.5 | 24.3 | 27.4 |
| Moderate | 50% | 22.0 | 37.8 | 22.4 | 8.4 | 25.0 | 28.1 |
| Ours | 50% | 24.4 | 40.9 | 25.1 | 9.9 | 27.1 | 31.5 |
| Random | 10% | 25.4 | 42.5 | 26.7 | 10.7 | 27.7 | 32.1 |
| Moderate | 10% | 25.0 | 42.7 | 25.9 | 11.2 | 28.5 | 33.0 |
| Ours | 10% | 26.7 | 44.0 | 27.8 | 12.1 | 30.0 | 35.1 |

# I    LIMITATION AND BROADER IMPACTS

We propose a novel efficient quantization-aware training method with coreset selection in this work and evaluate this method on various networks, datasets, and quantization settings. However, there are still limitations to the proposed method. As our analysis is based on the assumption of the classification task with SGD optimizer, its effectiveness on regression and other optimizers is not guaranteed. In addition, we did not evaluate the effectiveness and efficiency of our method on emerging deep learning models (for example, Transformers Vaswani et al. (2017) based language models) and other hardware systems.

As the first work exploring the data efficiency of quantization-aware training from the perspective of coreset selection, we believe our method can help improve the efficiency of QAT to help more researchers acquire quantized models in a shorter time. We also hope to reduce the energy cost ($CO_2$ emission) of expensive quantization-aware training. As we will make our codes and models publicly available, our work provides a solid baseline for future research related to quantization.

