# OpenReview forum: "Efficient Quantization-aware Training with Adaptive Coreset Selection"
_ICLR.cc/2024/Conference — Submitted to ICLR 2024_

### Official Review · Reviewer_NUL5 · 2023-10-29

**Soundness:** 3 good
**Presentation:** 3 good
**Contribution:** 3 good
**Rating:** 6
**Confidence:** 4

**Summary:**

The authors proposed a coreset selection pipeline for quantization-aware training in order to reduce the training cost. The authors introduced the error vector scores to measure the caused loss degradation if one data sample was removed. The authors further proved that knowledge distillation can benefit the coreset selection. The authors conducted both theoretical analysis and experimental results to show the efficacy of the method.

**Strengths:**

- The authors provided a new angle in reducing the training cost during quantization-aware training, i.e., using fewer training data. It was well-known that quantization-aware training was often time-consuming in comparison to post-training quantization. Reducing the training cost by using fewer samples is relatively novel and may benefit the following research.
- The authors provided a relatively solid theoretical proof to bound the caused training error when selecting a core subset of samples. This was especially encouraged.

**Weaknesses:**

- It would be great if the authors can provide the training cost comparison between this work and prior work. Since the main target was to reduce the training cost, just listing the ratio of used training samples may not be sufficient to justify the efficiency. I would like to see the real computation cost including core samples selection, KD, SGD training of the paper. I saw Tab.3 compared the training time, but it did not compared with other SOTA QATs.
- It seemed that this work can not achieve a lossless model in comparison to using the entire training samples. In the caption of Tab.1 and Tab.2, using full training set can almost always get a higher accuracy than using a subset, even if the ratio if 80%. If this was the case, we may not rely on coreset selection during QAT. Although the proposed methods always worked better than random selection, it was still important to show that the methods can reach the same accuracy level as using the full training set in QAT.

**Questions:**

See weaknesses

---

> ### Author Response · Authors · 2023-11-18
>
> Thanks for the supportive comments and detailed review. Here are our replies to your doubts and questions.
>
> **[Weakness 1: Please provide the training cost comparison between this work and prior work.]**
>
> We have provided the real training time comparison in Table 5 of the appendix. The results in Table 5 compare the whole training time with other coreset selection methods. Our method is efficient and brings minimal overhead to the standard SGD training with KD. We also provide the detailed training time composition as follows. Since KD will change the loss function of SGD training and is difficult to break down these two parts, we provide two sets of results w/ and w/o KD. The experiments are conducted on 2 NVIDIA RTX 3090 GPUs.
>
> | Strategy/Fraction    | 10%  | 30%  | 50%  | 60%  | 70%  | 80%  |
> |----------------------|------|------|------|------|------|------|
> | Ours (w/ KD)         | 11.3 | 20.7 | 36.1 | 42.2 | 48.0 | 53.8 |
> | Ours (w/ KD)-selection | 3.2  | 3.2  | 3.2  | 3.2  | 3.2  | 3.2  |
> | Ours (w/ KD)-training  | 8.1  | 17.5 | 32.9 | 39.0 | 44.8 | 50.6 |
> | Ours (w/o KD)        | 9.7  | 16.0 | 29.5 | 34.4 | 40.2 | 45.4 |
> | Ours (w/o KD)-selection | 3.2  | 3.2  | 3.2  | 3.2  | 3.2  | 3.2  |
> | Ours (w/o KD)-training  | 6.5  | 15.8 | 26.3 | 31.2 | 37.0 | 42.2 |
>
> The selection time is a constant and is only related to the full dataset size and the selection intervals $R$. The training with KD and without KD only influence the backpropagation time of the training and there is no backward in the selection phase, which makes the time of selection the same (~3.2h) across all settings.

---

> ### Author Response · Authors · 2023-11-18
>
> **[Weakness 2: It seemed that this work can not achieve a lossless model in comparison to using the entire training samples.]**
>
> Thanks for this great question. We would like to first clarify that **all existing coreset selection methods cannot achieve lossless acceleration** when the model size and training settings (training epochs, hyper-parameters, etc.) are appropriate. The target of coreset selection is to achieve an optimal trade-off between model performance and training efficiency. Previous coreset research [1,2,3,4] tries to minimize the accuracy decrease when training efficiency is improved.
>
> A theory for this accuracy-efficiency trade-off is the neural scaling law [5,6], where the test loss $L(D)$ of a model follows a power law to the dataset size $D$ as $L(D) = (D_c/D)^\alpha$. The $D_c$ is a constant and $\alpha \in (0,1)$ reflects that when the training data amount $D$ is reduced, the test loss $L(D)$ will always increase. What we do in this work is try to minimize $\alpha$ so that we can always significantly improve efficiency with only negligible accuracy loss in QAT.
>
> As mentioned in [5], when there is overfitting in the training, we can achieve lossless acceleration compared to full data training because the performance is not bounded by the data size, which is a possible case in QAT. In addition, we would like to highlight another application of our method: **identifying noisy labels**. The nature of advantage of our method is to select important training examples for QAT and remove those low-quality redundant examples. This is especially useful when there is label noise in the training set (some examples are not correctly labeled). This application is proposed in previous coreset research [7], but not verified on QAT. We follow the setting of [7] to experiment on the QAT of ResNet-18 on CIFAR-10 with 10% randomized labels (labels of 10% of training samples are randomly re-generated). The ResNet-18 is quantized to 2/32 for weights are activations. We train the quantized ResNet-18 for 200 epochs using a learning rate of 0.01, weight decay of 5e-4, batch size of 512, and SGD optimizer on one NVIDIA RTX 3090 GPU. The experiments are repeated 5 times with random seeds under different data fractions and the mean accuracy and standard deviation are reported. When full data are selected, the accuracy is 89.91$\pm$0.51%. The comparison of the QAT Top-1 accuracy (%) of 2/32-bit quantized ResNet-18 on CIFAR-10 with 10% randomized labels is shown as follow:
>
> | Method/Fraction | 10%              | 20%              | 30%              | 40%              | 50%              |
> |-----------------|------------------|------------------|------------------|------------------|------------------|
> | Random          | 83.14$\pm$0.51   | 84.77$\pm$0.80   | 84.82$\pm$0.57   | 85.09$\pm$0.15   | 85.33$\pm$0.60   |
> | EL2N Score [1]  | 85.80$\pm$0.51   | 86.02$\pm$0.55   | 87.18$\pm$0.35   | 87.51$\pm$0.29   | 88.01$\pm$0.45   |
> | Ours            | 88.31$\pm$0.27   | 89.53$\pm$0.62   | 89.95$\pm$0.39   | 90.21$\pm$0.39   | 90.23$\pm$0.18   |
>
> As can be seen from the results, our method outperforms all other selection baseline and even perform better than full-data training when coreset size $\mathcal{S}\geq30\%$. This shows that our method can successfully prune those samples with incorrect labels. In this case, we actually achieve a ''lossless acceleration'' as both the accuracy and efficiency are better than full-data training.
>
> - [1] Deep learning on a data diet: Finding important examples early in training, NeurIPS 2021
> - [2] Glister: Generalization based data subset selection for efficient and robust learning, AAAI 2021
> - [3] Retrieve: Coreset selection for efficient and robust semi-supervised learning, NeurIPS 2021
> - [4] Moderate coreset: A universal method of data selection for real-world data-efficient deep learning, ICLR 2023
> - [5] Scaling Laws for Neural Language Models, arxiv 2020
> - [6] Revisiting neural scaling laws in language and vision, NeurIPS 2022
> - [7] Coresets for Robust Training of Neural Networks against Noisy Labels, NeurIPS 2020

---

### Official Review · Reviewer_ALc7 · 2023-10-30

**Soundness:** 2 fair
**Presentation:** 2 fair
**Contribution:** 2 fair
**Rating:** 3
**Confidence:** 3

**Summary:**

This submission proposed to improve quantization-aware training (QAT) by using coreset, which contains the importance samples for better training. The coreset is generated by: 1) Estimation of loss brought by specific sample using first order expansion approximation, leading to gradient norm. 2) Disagreement score during distillation with a full-precision counterpart.

**Strengths:**

1. It is interesting to use data selection for QAT impovement.

**Weaknesses:**

1. The proposed methods (loss estimation and disagreement score) has no connection with QAT: these two algorithm can be applied to full-precision training without modification.
2. Gradient norm for sample important estimation is well knowed and widely used.

**Questions:**

I don't have questions currently. Please clarify whether the proposed method is related to QAT.

**Details Of Ethics Concerns:**

N.A.

---

> ### Author Response · Authors · 2023-11-18
>
> Thanks for the constructive comments and interest in our work. Please find our replies to your doubts and questions as follows.
>
> **[Weakness 1:The proposed methods (loss estimation and disagreement score) has no connection with QAT: these two algorithm can be applied to full-precision training without modification ]**
>
> We would like to clarify that the analysis of the loss estimation and disagreement score is **specifically designed** for quantization-aware training (QAT) and cannot be applied to full-precision training. The major difference between QAT and full-precision training is that the real-value weights $w^r$ of full-precision training are quantized to $b$-bit low-precision representation $w^q=q_b(w^r)$ during QAT as shown in section 3.1 of the paper. The quantization of weights results in a different activation and different output of the model, the quantized model output is $p(w^q,x)$ for a given input sample $x$, which is very different from $p(w^r,x)$ for the full-precision training.
>
> For the proposed disagreement score, its intuition is to measure the distance of full-precision model output $p(w^r,x)$ and quantized model output $p(w^q,x)$. For those samples with a higher disagreement score, it should be retained in the coreset as these samples are ``disagreed’’ by the quantized model and full-precision model. Since we use knowledge distillation in our settings, the full-precision teacher model prediction $p_\mathbf{T}(w^r,x)$ is utilized as a more accurate alternative of naive full-precision model output $p(w^r,x)$. For the full-precision training, there is **only one model prediction** $p(w^r,x)$ and we **cannot derive any distance** similar to the proposed disagreement score.
>
> In the analysis of loss estimation, all the gradient are computed on the quantized weights $w^q$, such as the gradient of loss on quantized weights and sample $(x_j, y_j)$ are $\frac{d \mathcal{L}(p(w^q_t,x_j), y_j)}{d w^q_t}$, and the gradient of quantized weights on iteration $t$ are $\frac{d w^q_t}{d t}$. This analysis only holds for the quantized model and cannot directly transfer to full-precision training.
>
> To sum up, the proposed method is fully quantization-aware and all the preconditions of the analysis on loss estimation and gradient approximation only hold for QAT settings. If the training is full-precision, we cannot compute the proposed metric as there is no quantized model prediction $p(w^q,x)$.
>
>
> **[Weakness 2: Gradient norm for sample important estimation is well known and widely used. ]**
>
> We also agree that gradient-norm methods are one of the most widely used coreset selection methods. We have discussed the most related gradient-norm-based methods in section 2 and give a detailed introduction of the GraNd-score method in section A of the appendix. However, we would like to clarify that **the proposed error vector score and disagreement score are highly different from previous gradient-norm-based methods** as follows:
>
> - One assumption from the previous gradient-norm-based method (GraNd-score [1]) is that this approximation only holds when the model has been trained for a few epochs, which requires us to perform early training on the model to have statistics to compute the score. However, data selection with our metrics (EVS and DS) does not perform any early training and all the proposed metrics are computed during the normal QAT phase, which significantly improves the efficiency of coreset selection.
>
> - In addition, storing all the gradients of the gradient-norm-based method incurs significant memory overheads during QAT. The efficiency will be lower than full dataset training with a high subset fraction. The performance with these metrics is sub-optimal as the converged quantized model in the later training epochs of QAT is not considered. We start from gradient analysis in our method, but do not need to compute the gradient for our proposed metric so there is no overhead for training.
>
>
> To summarize, our work is inspired by gradient-norm-based methods, but significantly different from them as we fully consider the quantization settings.
>
> [1] Deep learning on a data diet: Finding important examples early in training, NeurIPS 2021

---

> > ### Comment · Reviewer_ALc7 · 2023-11-23
> > **Response to Author**
> >
> > Thanks for your detailed response.
> >
> > For Weakness 1's disagreement score, full-precision setting can simply use another extra large model as teacher model. There **can be more than one model prediction**. As the full-precision teacher model is introduced in QAT setting, a larger full-precision model can be introduced in full-preicison setting.
> >
> > My point is that, the **distillation framework** is not designed specifically for QAT.
> >
> > For Weakness 1's loss estimation, **methodologically**, what is the difference between gradient of quantized weights and full-precision weights?
> >
> > Besides, is it typo in rebuttal that: "the gradient of quantized weights on iteration are $\frac{d w^q_t}{dt}$".
> >
> > More specifically, considering performing loss estimation on a full-precision models, the gradient of weights is still necessary and effective, just as it is used in QAT setting. **Setting does not change the usage of gradient**.  Author mentioned that "all the preconditions of the analysis on loss estimation and gradient approximation only hold for QAT settings", please let me know line number of the exact analysis in the main content if possible.
> >
> > I am wondering whether my understanding is correct.
> >
> > For Weakness 2, the connection between error vector score / disagreement score and QAT setting is challenged above and it is not fully addressed.
> >
> > It is of course different from gradient-norm-based methods. But my main concerns still lie: 1) The gradient norm for sample selection should not be considered contribution. 2) The inspired idea (from the idea of gradient norm) of error vector score / disagreement score is not novel as they are not connected to QAT setting.

---

> ### Author Response · Authors · 2023-11-23
>
> Thanks for your insightful questions again. We would like to further reply to your doubts regarding the connection of our method to QAT and gradient-norm-bsed methods.
>
> We first want to emphasize that the analysis in section 3.3 based on VC theory are fully **quantization-aware**. It is only because **QAT are more non-separable and difficult than full-precision training**, we can assume the value of $\alpha_q$ and  $\alpha_r$ to derive the inequality Eq.14 to show that we can select fewer data to achieve the same error upper bound in QAT.
>
> We then would like to highlight that selection with disagreement score in QAT does not require any **extra large model as teacher model** because the full-precision model can be the teacher of the same quantized model (e.g., full-precision ResNet-18 can be the teacher of quantized ResNet-18). However, in the full-precision training, there are **not more than one model prediction from a same model**.
>
> We agree that distillation framework is not designed specifically for QAT, but we would also like to first highlight that knowledge distillation is a widely-used and ‘default’ setting for previous quantization work[1-5] as we discussed in section F of the appendix, including the LSQ quantization [2] we use in this paper. This means that **our selection metric (disagreement score) can be a plug-and-play for most QAT methods**. However, for full-precision training, knowledge distillation is not a ‘default’ setting. Thus, we can not use a similar metric for selection in most cases.
>
> We still insist that **setting changes the usage of gradient**. Methodologically, If we replace the gradient of quantized weights by full-precision weights, the analysis will be very different. There is no need to give Eq.4 based on STE. Then there should not be any expected value or norm in Eq.5 because STE of quantization is not considered (STE only holds for weights within clipping range $-Q_N \leq x^r/s \leq Q_P$ as shown in Eq.2)
> In addition, the chain rule in Eq.6 will shrink to
> $\left.\frac{d \mathcal{L}}{d t}\right|_{(x_j, y_j),\mathcal{T}} = \frac{d \mathcal{L}(p(w^r_t,x_j), y_j)}{d w^r_t} \frac{d w^r_t}{dt}$.
>
> Then, there should also not be any norm $||\cdot||$ Eq.7 if we replace the $\frac{d \mathcal{L}}{d w^q_t}$ by $\frac{d \mathcal{L}}{d w^r_t}$. Since there is no randomness because all weights follow the same gradient characteristic in full-precision network, the Eq.8 will not hold because we cannot dereive any expectation over random minibatch sequence and the proposed $d_{\text{EVS}}$ cannot be used for selection.
>
> Thanks for pointing our the typo, in the rebuttal, we want to say "the continuous-time dynamic of quantized weights $w^q_t$ on iteration $t$ are $\frac{d w^q_t}{d t}$".
>
> For the concerns regarding the connection to gradient-norm-based methods, we have not claim the gradient norm as our contribution. The contribution is to **theoretically analyze how and why the proposed metrics can be used in the coreset selection for QAT**. As we have discussed how the proposed metric is connected to QAT setting (both for the gradient analysis and VC-theory based distillation analysis), the concern 2 should also be addressed.
>
> We sincerely hope our responses to your questions and concerns can help you understand how our method is connected to QAT. We thank you again for your time and efforts and do not hesitate to let us know if your have any additional comments on our responses.
>
> - [1] Apprentice: Using Knowledge Distillation Techniques To Improve Low-Precision Network Accuracy, ICLR 2018
> - [2] Learned Step Size Quantization, ICLR 2020
> - [3] Training Quantized Neural Networks with a Full-precision Auxiliary Module, CVPR 2020
> - [4] Nonuniform-to-Uniform Quantization: Towards Accurate Quantization via Generalized Straight-Through Estimation, CVPR 2022
> - [5] SDQ: Stochastic Differentiable Quantization with Mixed Precision, ICML 2022

---

> > ### Comment · Reviewer_ALc7 · 2023-11-23
> > **Response to Author**
> >
> > Thanks again for your clarification, here is what I have got after reading your response:
> >
> > 1. The theory is related to QAT setting. Properties of QAT contributes to a  same error upper bound compared with full-preicison counterpart.
> >
> > 2. Disagreement score can be regarded as a plug-an-play module for QAT methods with distillation, nevertheless, most QAT methods.
> >
> > 3. The usage of gradient on quantized weight is related to derivation of Eq.7 and corresponding error vector score.
> >
> > 4. Therotical analysis on application of proposed method in QAT  is the main contribution of the submission.
> >
> > And here is my concerns:
> >
> > 1. It seems the QAT is mainly related to the theory part. My first impression on the methods is that the submission borrows some common techniques in coreset selection in full-precision model training to QAT setting.
> > To be honest, I am not a fan of therotical analysis, especially the theory can not guide the invention of the algorithm or provide insights in the field. The theory in the submission is more like **providing analysis to the usage of error vector score / disagreement score in QAT**.
> > Though author may claim that the theory is very important and provides guarantee for the application, it still seems add less contribution to the submission for the method is not put forward based on the theory. Besides, the method is not innovative enough (as it can be applied in full-precision setting with minor modifications) without the theory.
> >
> > 2. Overall, my current understand of the submission is that: it provides two methods (error vector score / disagreement score) to improve the performance of QAT, which is analyzed and provided somehow guarantee of the applications.
> > Author claims that the **connection between the method and QAT lies in the theory** and the theory is a contribution. I agree that the theory somehow relates the method with QAT. But my concern is that the theory is an analysis and does not necessarily lead to the methods. Methods come first and then the theory is proposed for novelty.
> >
> > I am aware of the points in the author's response and parts of it convince me of the previous concerns on the connection problem.
> >
> > Please let me know if there is any misunderstanding.

---

> ### Author Response · Authors · 2023-11-23
>
> Thanks for your timely response and summarization. I would like to furthur provide more information to address your concern regarding the connection of the proposed method to QAT (include but not limited to therotical analysis). For the 2 concerns you raised in the reply, our response are:
>
> 1. As this is **the first work to research on the data-efficiency of QAT from coreset perspective**, no researchers before has applied any full-precision coreset selection methods to QAT. We are the first to conduct experiments shown in Table 1,2 to demonstrate that although full-precision coreset methods can be directly applied to QAT without any modification, their performance are sub-optimal (some of them even cannot outperform random selection in QAT). While these methods can help full-precision training, they fail to improve QAT effectively. We believe the core reason of the failure of application lie in the underlying difference between QAT and full-precision training. Then we start from the gradient/loss/error bound analysis and the difference between the gradient/loss/error bound characteristic leads me to propose these two metrics. The **performance difference** of previous methods on QAT and full-precision training lead us to analyze the **difference of gradient/loss/error**, and these analysis on difference **lead us to propose these two metrics**. We believe that our method are very different from previous full-preicison counterpart as our methods is proposed based on QAT characteristics and verified on QAT experiments. It is unfair to underestimated the novelty only because we use a norm of gradient in our method.
>
> 2. In this research, we actually start from experiments of directly applying full-precision coreset methods to QAT, then we conduct analysis based on the results of experiments, and proposed the two metrics in the final stage. For example, it is because we find that there are randomness in gradient that some weights (within clipping range) have different gradient compared to other, we can then compute the norm of gap between the label and quantized prediction. It is because we find that QAT usually have a higher loss and lower accuracy compared to full-precision training, we can then analyze the error bound to find that teacher model's prediction is a good target for the selection metrics to maximize the data efficiency. We believe that the **experiments, analysis, and the proposed method** are highly novel.
>
> We believe that our discussion are highly constructive to better demonstrate the novelty of this submission. More analysis beyond theoretical proof will be added in our revision to better show the connection of our method to QAT. We thank you again for your time and efforts and do not hesitate to let us know if your have any additional comments on our responses. If you feel your concerns have been addressed, please kindly consider if it is possible to update your rating.

---

### Official Review · Reviewer_iCtg · 2023-10-31

**Soundness:** 2 fair
**Presentation:** 3 good
**Contribution:** 1 poor
**Rating:** 3
**Confidence:** 5

**Summary:**

The paper presents an approach to enhance the efficiency of Quantization-Aware Training (QAT) using coreset selection. The authors examine the redundancy present in the training data of QAT and determine that significance varies among samples. To quantify the importance of each sample for QAT, they develop two novel metrics: Error Vector Score (EVS) and Disagreement Score (DS). These metrics are derived through theoretical analysis of the loss gradient. Subsequently, the authors introduce a groundbreaking approach called Quantization-Aware Adaptive Coreset Selection (ACS) that adaptively chooses samples that are informative and relevant to the current training stages, taking into account the EVS and DS metrics. The proposed ACS has been rigorously tested across various network architectures, datasets, and quantization settings, and the results demonstrate that ACS can improve training efficiency while maintaining the overall performance.

**Strengths:**

1.The utilization of coreset selection to enhance the efficiency of quantization-aware training (QAT) constitutes a pioneering method that has not been thoroughly investigated previously, making it an contribution to the field.
2.The paper is written in a clear and concise manner, featuring a succinct yet comprehensive explanation of the proposed approach and the associated metrics. The authors have also provided a detailed description of the experimental setup, allowing readers to replicate their experiments and compare their results with other methods.

**Weaknesses:**

1. There are no advantages from the experimental results in Table 1 and 2. The Top-1 accuracy of 80% selected data is worse than full-data QAT and the relative training time is longer than the full-data training(53.8/0.8=67.25 > 62.3h).
2. In Table 4, the best strategy for different ratio are various, which make the method not robust and universal.

**Questions:**

1.Could you please provide the experimental results (accuracy and training time) without KD?
2.What's the difference between the proposed EVS and Shapley values?
3.Do the preconditions of the simplification and approximation meet by the ordinary QAT networks?

---

> ### Author Response · Authors · 2023-11-18
>
> Thanks for the constructive feedback and detailed comments. Attached are our replies to your doubts and questions.
>
> **[Weakness 1: There are no advantages from the experimental results in Table 1 and 2.]**
>
> Thanks for this great question. We would like to first clarify that **all existing coreset selection methods cannot achieve an advantage over full data training** when the mode size and training settings (training epochs, hyper-parameters, etc.) are appropriate under ideal cases. However, in real-world settings, some labels… In addition, the **real** training time with 80% coreset is 53.8h, which is shorter than full data training (62.3h). The relative training time makes no sense in the setting of coreset selection and is never discussed in previous research.
>
> **In an ideal case**, the real advantage of coreset selection methods is to achieve an optimal trade-off between model performance and training efficiency. Previous coreset research [1,2,3,4] tries to minimize the accuracy decrease when training efficiency is improved. A theory for this accuracy-efficiency trade-off is the neural scaling law [5,6], where the test loss $L(D)$ of a model follows a power law to the dataset size $D$ as $L(D) = (D_c/D)^\alpha$. The $D_c$ is a constant and $\alpha \in (0,1)$ reflects that when the training data amount $D$ is reduced, the test loss $L(D)$ will always increase. What we do in this work is try to minimize $\alpha$ so that we can always significantly improve efficiency with only negligible accuracy loss in QAT.
>
> **In real-world scenarios**, as mentioned in [5], when there is overfitting in the training, we can achieve lossless acceleration compared to full data training because the performance is not bounded by the data size, which is a highly possible case in QAT. In addition, we would like to highlight another application of our method: **identifying noisy labels**. The nature of advantage of our method is to select important training examples for QAT and remove those low-quality redundant examples. This is especially useful when there is label noise in the training set (some examples are not correctly labeled). This application is proposed in previous coreset research [7], but not verified on QAT. We follow the setting of [7] to experiment on the QAT of ResNet-18 on CIFAR-10 with 10% randomized labels (labels of 10% of training samples are randomly re-generated). The ResNet-18 is quantized to 2/32 for weights are activations. We train the quantized ResNet-18 for 200 epochs using a learning rate of 0.01, weight decay of 5e-4, batch size of 512, and SGD optimizer on one NVIDIA RTX 3090 GPU. The experiments are repeated 5 times with random seeds under different data fractions and the mean accuracy and standard deviation are reported. When full data are selected, the accuracy is 89.91$\pm$0.51%. The comparison of the QAT Top-1 accuracy (%) of 2/32-bit quantized ResNet-18 on CIFAR-10 with 10% randomized labels is shown as follow:
>
> | Method/Fraction | 10%              | 20%              | 30%              | 40%              | 50%              |
> |-----------------|------------------|------------------|------------------|------------------|------------------|
> | Random          | 83.14$\pm$0.51   | 84.77$\pm$0.80   | 84.82$\pm$0.57   | 85.09$\pm$0.15   | 85.33$\pm$0.60   |
> | EL2N Score [1]  | 85.80$\pm$0.51   | 86.02$\pm$0.55   | 87.18$\pm$0.35   | 87.51$\pm$0.29   | 88.01$\pm$0.45   |
> | Ours            | 88.31$\pm$0.27   | 89.53$\pm$0.62   | 89.95$\pm$0.39   | 90.21$\pm$0.39   | 90.23$\pm$0.18   |
>
> As can be seen from the results, our method outperforms all other selection baseline and even perform better than full-data training when coreset size $\mathcal{S}\geq30$%. This shows that our method can successfully prune those samples with incorrect labels. In this case, we actually achieve a ''lossless acceleration'' as both the accuracy and efficiency are better than full-data training.
>
> - [1] Deep learning on a data diet: Finding important examples early in training, NeurIPS 2021
> - [2] Glister: Generalization based data subset selection for efficient and robust learning, AAAI 2021
> - [3] Retrieve: Coreset selection for efficient and robust semi-supervised learning, NeurIPS 2021
> - [4] Moderate coreset: A universal method of data selection for real-world data-efficient deep learning, ICLR 2023
> - [5] Scaling Laws for Neural Language Models, arxiv 2020
> - [6] Revisiting neural scaling laws in language and vision, NeurIPS 2022
> - [7] Coresets for Robust Training of Neural Networks against Noisy Labels, NeurIPS 2020

---

> ### Author Response · Authors · 2023-11-18
>
> **[Weakness 2: In Table 4, the best strategy for different ratio are various, which make the method not robust and universal.]**
>
> Thanks for mentioning this question. Across all data fractions (including the 30% ratio setting), selection with the cosine annealing strategy yields top-2 accuracy. In the 30% ratio setting, the performance difference between linear strategy (71.11%) and cosine strategy (71.09%) is negligible. Since the cosine annealing strategy outperforms the linear strategy across all other ratios, we utilize cosine as our final strategy. We provide the analysis results of the annealing strategy of MobileNet-V2 on the CIFAR-100 dataset as follows. The training settings are the same as we mentioned in the paper (We also repeat each experiment 5 times with random seeds, and only report mean accuracy for a direct comparison of different strategies).
>
> | Strategy/Fraction | 10%   | 20%   | 30%   | 40%   | 50%   | Average |
> |------------------|-------|-------|-------|-------|-------|---------|
> | linear           | 63.55 | 65.78 | 66.40 | 66.79 | 67.08 | 65.92   |
> | cosine           | 63.67 | 65.91 | 66.41 | 66.85 | 67.19 | 66.01   |
>
> As can be seen from the results, the optimal strategy across all fractions is also cosine annealing. The trend on CIFAR-100 aligns with our results on ImageNet-1K, which makes our method robust and universal.
>
> **[Question 1: Please Provide the experimental results (accuracy and training time) without KD.]**
>
> Thanks for the question, we would like to first highlight that KD is applied to all experiments regardless of the precision and dataset fraction. The full-data training baseline also involves training with knowledge distillation. KD is a normal and ‘default’ setting for all previous quantization work[1-5], including the LSQ quantization [2] we use in this paper. For a fair comparison with previous work, we equally utilize KD for all the experiments in this work.
>
> Based on your requirements, we remove the knowledge distillation in our method. Since the DS is built on the soft label of the teacher model, we do not use it in our selection w/o KD. Only EVS is applied as the selection metric. We follow the same settings for quantized ResNet-18 on ImageNet-1K as shown in our paper. The training time and accuracy are shown as follows. When full data are selected ($S=100%$), the Top-1 accuracy is** 70.21%** and the training time is **53.1h**. From the results, we can see that our method still outperforms previous SOTA without KD and the training efficiency is still optimal.
>
> | Strategy/Fraction     | 10%   | 30%   | 50%   | 60%   | 70%   | 80%   |
> |-----------------------|-------|-------|-------|-------|-------|-------|
> | Ours (w/o KD)         | 66.91 | 68.77 | 69.25 | 69.66 | 69.85 | 70.03 |
> | EL2N (w/o KD)         | 59.71 | 63.50 | 65.19 | 66.38 | 67.90 | 69.01 |
> | Glister (w/o KD)      | 62.41 | 65.18 | 66.47 | 67.05 | 68.81 | 69.45 |
> | Ours (w/o KD) training time | 9.7h  | 16.0h | 29.5h | 34.4h | 40.2h | 45.4h |
> | EL2N (w/o KD) training time | 10.6h | 17.1h | 31.3h | 35.0h | 42.9h | 48.0h |
> | Glister (w/o KD) training time | 11.9h | 22.7h | 34.5h | 41.7h | 50.3h | 56.9h |
>
> **[Question 2: What’s the Difference between the proposed EVS and Shapley values?]**
>
> If the Shapley values you mentioned are the Shapley values for feature selection [6,7], we would like to clarify that it’s different from our EVS from the following perspectives:
> - Shapley value is designed for **feature selection**, which assumes the dataset is fixed. However, the EVS is designed for **coreset selection**, which assumes the features from our model are fixed and there are no choices of different features.
> - Shapley value is derived from the weighted average over all contributions by the **features**. While EVS is specifically designed for QAT and is derived from the expected gradient norm on a specific training **sample**.
> - Shapley value is **static**. Given the feature set ${i}$, a set of objects $F$ of a Transferable Utility (TU) game, and the evaluation metric $C(F)$, we can compute the Shapley value without any training. However, the EVS is **dynamic**, which is computed during the training process of QAT and we will adaptively select the coreset based on it.
>
> These discussions have been included in the related work of our revision.
>
> - [1] Apprentice: Using Knowledge Distillation Techniques To Improve Low-Precision Network Accuracy, ICLR 2018
> - [2] Learned Step Size Quantization, ICLR 2020
> - [3] Training Quantized Neural Networks with a Full-precision Auxiliary Module, CVPR 2020
> - [4] Nonuniform-to-Uniform Quantization: Towards Accurate Quantization via Generalized Straight-Through Estimation, CVPR 2022
> - [5] SDQ: Stochastic Differentiable Quantization with Mixed Precision, ICML 2022
> - [6] Feature selection based on the shapley value, IJCAI 2005
> - [7] Shapley values for feature selection: The good, the bad, and the axioms, arxiv 2021

---

> ### Author Response · Authors · 2023-11-18
>
> **[Question 3: Do the preconditions of the simplification and approximation meet by the ordinary QAT networks?]**
>
> We apologize that we do not clearly understand what is ordinary QAT networks. All the analyses in this paper including the simplification and approximation are based on the QAT settings.
>
> If you mean a full-precision network, the preconditions are not met since the gradient is derived from the quantized weights and activations. This is why our methods are highly quantization-aware.
> If you mean a quantized network without KD, the preconditions of the gradient are met (all the analyses in sections 3.1 and 3.2) and we can still use the error vector score (EVS) for the selection. The VC-dimension analysis is based on the soft label of KD, which makes the disagreement score (DS) not applicable. In this case, our method still can outperform all previous SOTA as shown in the reply to question 1.
>
> If you can provide more details about “**ordinary QAT networks**”, we are always pleased to answer your follow-up questions.

---

### Author Response · Authors · 2023-11-18
**General response**

We appreciate all reviewers for your effort to review our paper. We are encouraged by the positive comments, such as applying coreset selection to improve the efficiency of quantization-aware training (QAT) is novel and interesting [Reviewer iCtg, ALc7, NUL5], the theoretical proof of the training error on coreset is solid [Reviewer NUL5], the paper is well-written and detailed experimental setup are provided [Reviewer iCtg], the proposed approach effectively accelerating the QAT [Reviewer NUL5], etc.

We also thank the reviewers for constructive comments, such as clarifying how the proposed method is related to QAT[Reviewer ALc7], providing the experimental results without KD [Reviewer iCtg], the training cost comparison between this work and previous methods [Reviewer NUL5]. These suggestions will help us better demonstrate the effectiveness of the proposed ACS. We will accommodate all of the comments in our revision.

We would like to highlight the following modifications we made in the revised draft (the revised and appended content are shown in red color in the draft):
- We add the discussion over why our method cannot achieve lossless acceleration compared to full-data training in ideal cases. Theoretical analysis with the neural scaling law and experiments on the dataset with label noise are provided to prove the superiority of our method in real-world scenarios.
- We add more detailed experimental results with and without knowledge distillation (KD). Both the accuracy and real training time are shown. We also discussed and emphasized that KD is a normal and ‘default’ setting for all previous quantization work.
- We add Shapley values in the related work discussion.
- We add more discussion to highlight how our method is quantization-aware and the proposed analysis and metric cannot be applied to full-precision training.

We provide detailed responses to each reviewer’s questions and concerns in the following. We thank you again for all the reviewers' time and efforts! Please do not hesitate to let us know of any additional comments on our responses.

---

### Meta-Review · Area_Chair_PMjm · 2023-12-16

**Metareview:**

The paper presents a method for speeding training of quantized neural networks by automatically selecting a training data curriculum.  The selection criteria are based on the expected loss due to removing a sample as well as disagreement with a teacher network from which the quantized student network is distilled.

Reviewer opinion is mixed, with Reviewer NUL5 giving a marginal accept and Reviewers ALc7 and iCtg favoring reject.  From the reviews and subsequent discussion, the AC agrees with some aspects of both Reviewer NUL5 and Reviewer ALc7's comments.  The paper does make a contribution to quantized training methodology, but the degree of technical innovation is somewhat unclear, as the approach appears to primarily be an application of existing coreset selection methods.  Reviewer ALc7 is concerned that "the distillation framework is not designed specifically for QAT" and "the method is not innovative enough (as it can be applied in full-precision setting with minor modifications)."  The AC believes additional work is needed to clarify the methodological innovation as well as its scope.

**Justification For Why Not Higher Score:**

Unresolved concerns about overall methodological novelty and scope.

**Justification For Why Not Lower Score:**

N/A

---

### Decision · Program_Chairs · 2024-01-16

Reject